# Programmed conversion of hypertrophic chondrocytes into osteoblasts and marrow adipocytes within zebrafish bones

Dion Giovannone[†], Sandeep Paul[†], Simone Schindler, Claire Arata, D'Juan T Farmer, Punam Patel, Joanna Smeeton, J Gage Crump*

Department of Stem Cell Biology and Regenerative Medicine, Keck School of Medicine, University of Southern California, Los Angeles, United States

**Abstract** Much of the vertebrate skeleton develops from cartilage templates that are progressively remodeled into bone. Lineage tracing studies in mouse suggest that chondrocytes within these templates persist and become osteoblasts, yet the underlying mechanisms of this process and whether chondrocytes can generate other derivatives remain unclear. We find that zebrafish cartilages undergo extensive remodeling and vascularization during juvenile stages to generate fat-filled bones. Growth plate chondrocytes marked by *sox10* and *col2a1a* contribute to osteoblasts, marrow adipocytes, and mesenchymal cells within adult bones. At the edge of the hypertrophic zone, chondrocytes re-enter the cell cycle and express *leptin receptor* (*lepr*), suggesting conversion into progenitors. Further, mutation of *matrix metalloproteinase 9* (*mmp9*) results in delayed growth plate remodeling and fewer marrow adipocytes. Our data support Mmp9-dependent growth plate remodeling and conversion of chondrocytes into osteoblasts and marrow adipocytes as conserved features of bony vertebrates.
DOI: https://doi.org/10.7554/eLife.42736.001

**\*For correspondence:**
gcrump@usc.edu

[†]These authors contributed equally to this work

**Competing interests:** The authors declare that no competing interests exist.

## Introduction

Vertebrate bones develop via two largely distinct processes. Intramembranous (i.e. dermal) bone, which makes up a large portion of the skull, arises through the direct differentiation of mesenchymal precursors into osteoblasts and then osteocytes. In contrast, endochondral bone, which comprises the majority of the axial and limb skeletons, arises through the progressive remodeling of an embryonic cartilage template. On the outside of developing endochondral bone, perichondral cells mature into periosteal progenitors that contribute to the bone collar. The cartilage templates of endochondral bone are organized into distinct zones of chondrocytes: resting, proliferative, pre-hypertrophic, and hypertrophic. In mammals, chondrocytes at the edge of the developing hypertrophic zone largely disappear as the cartilage matrix is degraded, a process concurrent with the invasion of blood vessels, hematopoietic cells, and progenitors for osteoblasts and marrow adipocytes (*Maes et al., 2010*). This growth plate remodeling contributes to the establishment of trabecular bone, complementing the cortical bone largely derived from the periosteum, and the marrow cavity supports continued hematopoiesis. As with mammals, zebrafish also have intramembranous and endochondral bones. Their endochondral bones are hollow and filled predominantly with fat yet do not support hematopoiesis as in mammals (*Witten and Huysseune, 2009*; *Weigele and Franz-Odendaal, 2016*). It has remained unclear, however, whether zebrafish bone arises solely through osteoblast differentiation in the periosteum, or also through invasion of the vasculature and

**eLife digest** Our adult bones are made of a fatty tissue, called marrow, wrapped inside a hard outer layer produced by bone cells. They may appear stiff and unyielding, but our bones are actually dynamic structures. Early in life, most bones start as small 'templates' made of another, flexible tissue called cartilage. As the templates grow into adult bones, the cartilage is gradually replaced by bone and fat, but this process is still poorly understood. For example, it is not clear whether cartilage cells simply die and make way for new cells, or instead if they turn into bone and fat cells. To investigate this question, Giovannone, Paul et al. set out to follow the fate of early cartilage cells in zebrafish, and to compare this with what happens in mammals. Zebrafish were chosen because their skeleton and ours develop in similar ways; yet, these animals are much easier to study, in particular because their embryos are transparent.

Young cartilage cells were 'tagged' with a long-lasting fluorescent protein in genetically engineered zebrafish embryos, and then followed over time. As the embryos started to form bones, the cartilage cells gave rise to bone cells, fat cells, and also potentially adult stem cells within the marrow, which can become other types of cells. This process required a protein called Mmp9, which also helps shape bone development in other organisms, including humans.

Defects in how early cartilage templates morph into bone and fat may contribute to dwarfism and other severe conditions. Fully grasping the molecular mechanisms that preside over this complex transition may one day help design drugs to treat skeletal disorders.
DOI: https://doi.org/10.7554/eLife.42736.002

conversion of growth plate cartilage to bone as in mammals. The source of marrow adipocytes also remains unclear in either fish or mammals.

It has long been appreciated that many hypertrophic chondrocytes undergo cell death during endochondral ossification, with osteoblasts forming from periosteal cells brought into the bone along with the vasculature (*Maes et al., 2010*). At the same time, there are numerous studies showing that cultured chondrocytes can dedifferentiate into mesenchymal progenitors and/or transdifferentiate into osteoblasts (*Shimomura et al., 1975*; *Mayne et al., 1976*; *von der Mark and von der Mark, 1977*). Recent lineage tracing studies of hypertrophic chondrocytes, using constitutive and inducible *Col10a1-Cre-* and *Aggrecan-Cre*-based transgenes in mice, has revealed that such transdifferentiation may also occur in vivo, with chondrocytes making a major contribution to osteoblasts within trabecular bone and potentially also the bone collar (*Yang et al., 2014*; *Kobayashi et al., 2014*; *Jing et al., 2015*; *Park et al., 2015*). A limitation of these studies is the use of population-based labeling by Cre recombination, which cannot exclude low-level and/or leaky labeling of other cell types. It is also unclear whether hypertrophic chondrocytes can give rise to other cell types, such as marrow adipocytes, and whether hypertrophic chondrocytes directly transform into osteoblasts or do so through a stem cell intermediate. Finally, it is unknown whether the ability of chondrocytes to generate osteoblasts and other cell types is specific to mammals or a more broadly shared feature of vertebrates.

In this study, we address the long-term fate of growth plate chondrocytes in zebrafish, as well as potential mechanisms of their fate plasticity. We use the ceratohyal (Ch) bone of the lower face as a model. This long bone, which is derived from cranial neural crest cells, exhibits properties in common with the long bones of mammalian limbs, including two prominent growth plates at either end and a marrow cavity (*Paul et al., 2016*). Here, we describe remodeling of the Ch from a cartilage template to a fat-filled bone in juvenile stages, which coincides with extensive vascularization. Using inducible Cre and long-lived histone-mCherry fusion proteins, driven by regulatory regions of the chondrocyte genes *sox10* and *col2a1a*, we reveal contribution of chondrocytes to osteoblasts, adipocytes, and mesenchymal cells within the adult Ch. In mouse, *LepR* expression marks bone marrow cells that contribute to osteoblasts and adipocytes primarily after birth (*Zhou et al., 2014b*). In zebrafish, we find that growth plate chondrocytes express *lepr* and re-enter the cell cycle during the late hypertrophic phase, raising the possibility that *Lepr* +skeletal stem cells may derive from growth plate chondrocytes. Further, we find that delayed remodeling of the hypertrophic cartilage zone in zebrafish *mmp9* mutants correlates with a paucity of marrow adipocytes. Unlike in mouse where

Mmp9 functions in hematopoietic cells for timely growth plate remodeling (*Vu et al., 1998*), we find that Mmp9 is sufficient in neural crest-derived chondrocytes of zebrafish for growth plate remodeling. Our studies reveal that growth plate chondrocytes generate osteocytes and adipocytes in zebrafish bones, potentially by transitioning through a proliferative intermediate.

## Results

### Remodeling of the Ch bone in juvenile zebrafish

In order to characterize the progressive remodeling of an endochondral bone in zebrafish, we performed pentachrome staining on sections of the Ch bone from juvenile through adult stages (*Figure 1*). The Ch bone is shaped like a flattened barbell, and here we sectioned it to reveal the thin plane of the bone (see *Figure 1—figure supplement 1A*) for a view along the thicker perpendicular plane). Unlike the unidirectional growth plates in the mouse limb, the two growth plates of Ch are bidirectional with a central zone of compact, proliferative chondrocytes flanked by hypertrophic chondrocytes on either side (*Paul et al., 2016*). Unlike in many other fish species, the Ch bone, as with other bones in zebrafish, also contains embedded osteocytes (*Witten and Huysseune, 2009*). At 11 mm standard length (SL) (approx. 4.5 weeks post-fertilization (wpf)), the Ch contains chondrocytes throughout its length with the exception of a small marrow space at the anterior tip. The Ch is surrounded by a thin layer of cortical bone that has been shown to derive from osteoblasts located on the outside of the cartilage template (i.e. periosteum) (*Paul et al., 2016*). By 12 mm SL (approx. five wpf), both tips of the Ch contain marrow spaces, and on the central sides of the growth plates we begin to observe small fissures in the cortical bone and disruption of the hypertrophic zone. By 13 mm SL (approx. 5.5 wpf), breaks in the cortical bone become more prominent and are accompanied by further degradation of the cartilage matrix. At later stages (16 and 19 mm SL) (approx. 7 and 9 wpf), cortical bone regains integrity and increases in thickness, and marrow adipocytes containing LipidTOX +lipid vesicles are seen throughout Ch (*Figure 1—figure supplement 1B*). By adulthood (one year of age), the marrow cavity is filled with large fat cells and the growth plates appear largely mineralized. While we focus on the Ch for this study, a number of other cartilage-derived bones in the face and fins have been reported to have a similar structure in zebrafish, including growth plates and prominent marrow fat (*Weigele and Franz-Odendaal, 2016*).

### Vascularization of the Ch bone in juvenile zebrafish

Given the transient breakdown of cortical bone, we examined whether this coincides with vascularization of Ch. To do so, we performed confocal imaging of the dissected Ch from fish carrying both a chondrocyte-specific *col2a1a*:mCherry-NTR transgene and *fli1a*:GFP (*Figure 2A*). We used *fli1a*:GFP to label endothelial cells of the vasculature, but we also noticed that presumptive resting chondrocytes in the middle of the growth plate express this transgene. At 10 and 11 mm SL, vessels expressing *fli1a*:GFP are found largely on the outside of the Ch bone. By 13, 16, and 20 mm SL, we observe increasing numbers of capillaries within the Ch, coinciding with the replacement of *col2a1a*:mCherry-NTR +chondrocytes with adipocytes. High-magnification confocal sections at 18 mm SL clearly show *fli1a*:GFP+ vessels in intimate association with adipocytes and within the Calcein Blue+ bone collar (*Figure 1—figure supplement 1C*). Given the more complicated expression pattern of *fli1a*:GFP, we also independently confirmed blood vessel identity with *kdrl*:GFP. At 28 mm SL (approx. 26 wpf), the Ch is heavily supplied with both *kdrl*:GFP+blood vessels and *lyve1*:DsRed+lymphatic vessels, which abut each side of the growth plate (*Figure 2B*). Hence, as in mammalian bones, remodeling of the Ch bone is accompanied by extensive vascular invasion of the cartilage template.

### Contribution of *sox10*-lineage cells to osteoblasts, adipocytes, and mesenchymal cells

Given the extensive remodeling and vascularization of Ch, we next investigated the long-term fate of growth plate chondrocytes by multiple, independent methods. First, we constructed an inducible *sox10*:CreERT2 line and crossed it to a ubiquitous *bactin2*:loxP-tagBFP-stop-loxP-DsRed reporter (Blue to Red conversion: *B > R*). The zebrafish *sox10* promoter drives expression in early cranial neural crest cells from 10 to 16 hpf, followed by a second wave of expression in all chondrocytes from two dpf onwards (*Dutton et al., 2008*). Here, we took advantage of this second wave of expression

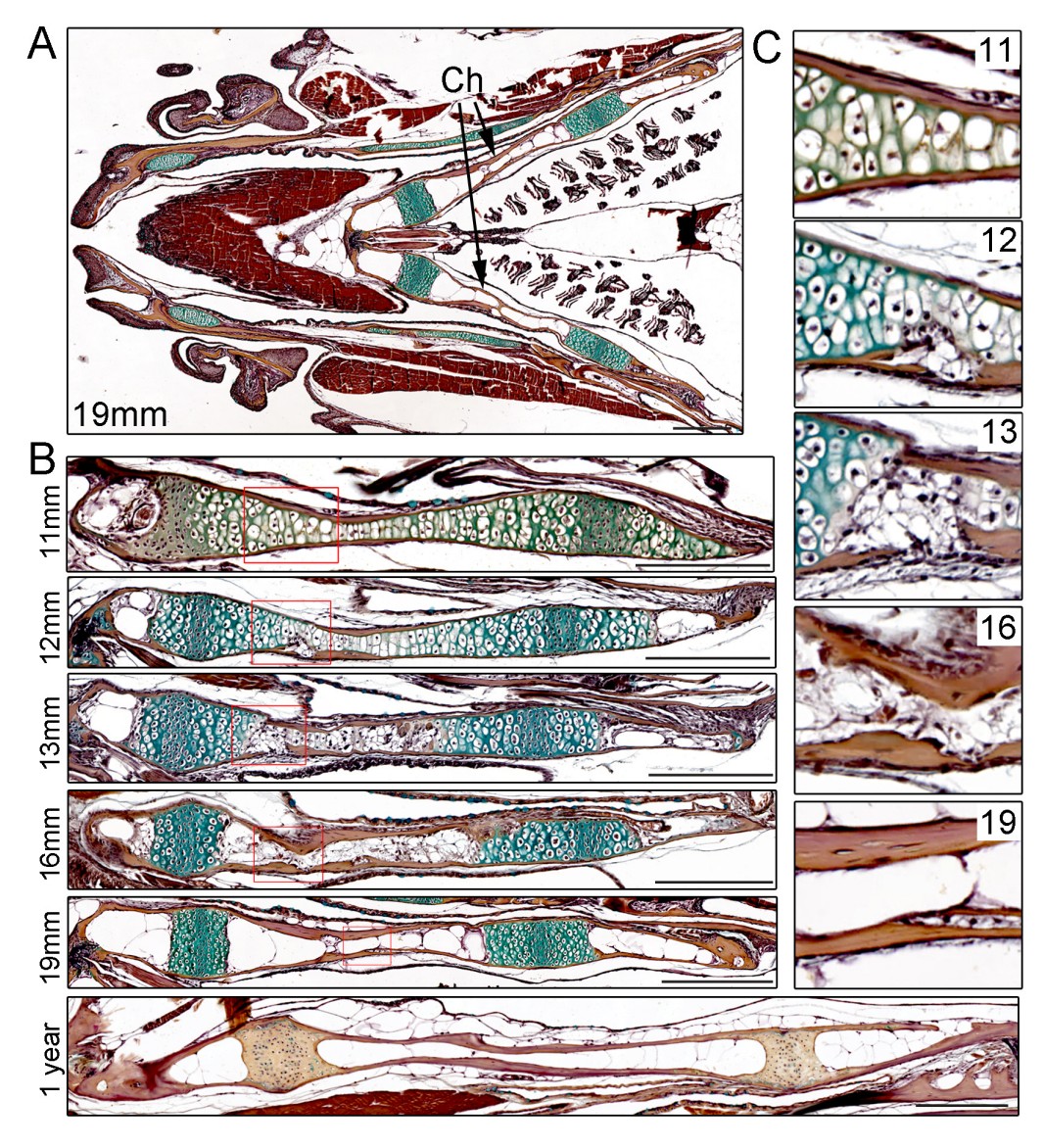

**Figure 1.** Time-course of Ch remodeling in juvenile zebrafish. (**A**) Pentachrome staining of a longitudinal section through the head of a 19 mm fish. The jaw is toward the left (anterior) and the gills toward the right (posterior). The green stain highlights the collagen matrix of cartilage, and the reddish-brown stain the mineralized matrix of bone. The bilateral set of Ch bones is indicated. $n = 3$. (**B**) High magnification views of the Ch at successive stages show the gradual replacement of chondrocytes in the central shaft and at each end with adipocytes (which appear white due to loss of lipid during processing). $n = 3$ for each stage. (**C**) Higher magnification views of the boxed regions in (**B**). Cortical bone appears reddish-brown. Note the breaks in cortical bone toward the lower part of the images at 12 and 13 mm, which are largely resolved by 16 and 19 mm. Scale bars = 50 μM.

DOI: https://doi.org/10.7554/eLife.42736.003

The following figure supplement is available for figure 1:

**Figure supplement 1.** Ch bone and marrow fat structure.

DOI: https://doi.org/10.7554/eLife.42736.004

to label developmental chondrocytes. Upon addition of 4-hydroxytamoxifen (4-OHT) at 15 dpf, we observed extensive labeling of chondrocytes within 5 days, as well as some cells in the perichondrium surrounding Ch and other cartilages (*Figure 3A*). We did not observe leaky conversion in the absence of 4-OHT at either embryonic or adult stages (*Figure 3—figure supplement 1A*). We then converted *sox10/B > R* fish by 4-OHT treatment at 14 dpf and raised these to adulthood (27 mm SL)

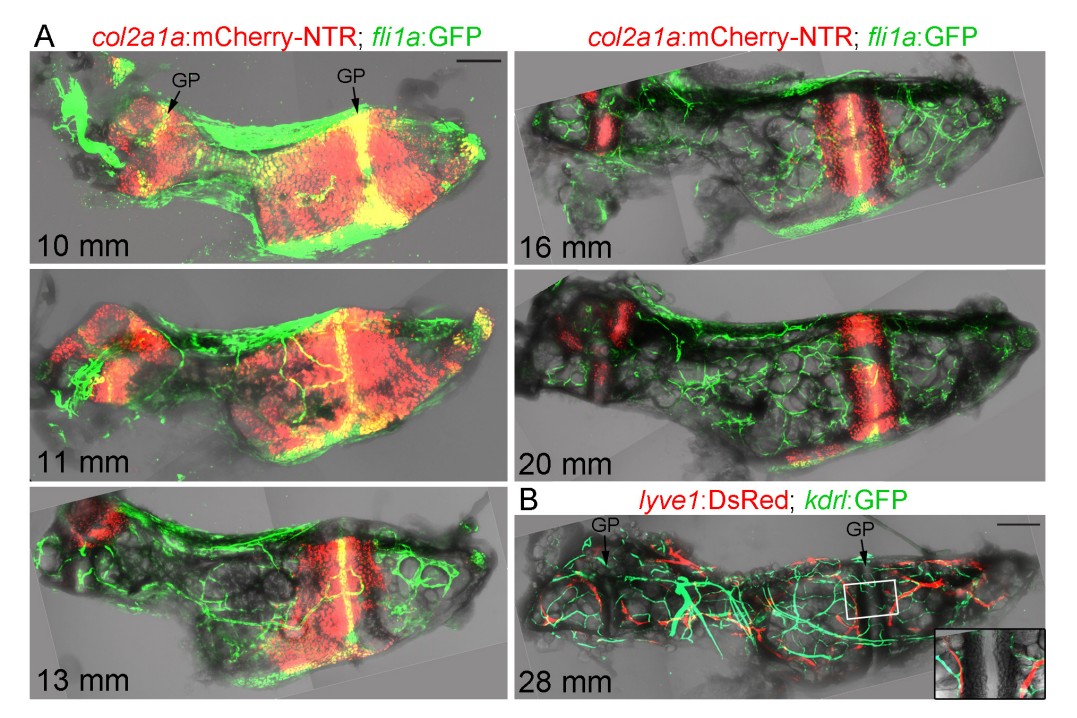

**Figure 2.** Vascularization of the Ch. (**A**) Confocal projections of dissected Ch bones at five successive stages. Merged fluorescent and brightfield channels show the gradual replacement of the cartilage with a fat-filled core. *col2a1a*:mCherry-NTR highlights chondrocytes that become increasingly restricted to two growth plates (GP) at either end of the bone. *fli1a*:GFP labels endothelial cells and chondrocytes located in the central portions of the growth plates. Vascularization of the Ch increases over time. *n* = 2 at each stage. (**B**) Confocal projection shows networks of *kdrl*:GFP+ vascular endothelial and *lyve1*:DsRed+ lymphatic endothelial cells within an adult Ch bone. The inset shows a single confocal section through the boxed portion of the growth plate, with both blood and lymphatic vessels abutting the edges but not penetrating into the growth plate. *n* = 2. Scale bars = 100 μm (**A**) and 200 μm (**B**).

DOI: https://doi.org/10.7554/eLife.42736.005

for analysis, with inclusion of an *ocn*:GFP transgene allowing us to label osteoblasts (*Figure 3B*). Confocal maximum intensity projections through Ch revealed extensive labeling of growth plate chondrocytes. We also observed numerous cells throughout the Ch, including in and around the marrow cavity. In sections through the middle of Ch, we observed labeling of a number of large diameter cells of adipocyte morphology (*Figure 3C*). In superficial sections through the cortical bone, we also observed labeled cells that were positive for the osteoblast marker *ocn*:GFP, as well as some labeled cells negative for *ocn*:GFP that may represent bone progenitors or other cell types (*Figure 3D*). As a comparison, we used a constitutive Cre driven by a human *SOX10* enhancer that drives expression throughout the neural crest lineage (note that this human enhancer lacks the second wave of chondrocyte expression seen with the zebrafish regulatory region used for the *sox10*: CreERT2 line) (*Kague et al., 2012*). When crossed to the *B > R* line, this neural crest-specific *SOX10*: Cre line drives broader conversion in the five dpf head than the later conversion of *sox10*:CreERT2 (*Figure 3—figure supplement 2A*). Analysis of the adult Ch in *SOX10*:Cre fish shows labeling of all growth plate cartilage, as well as most if not all adipocytes and numerous smaller cells throughout the marrow and cortical bone surface (*Figure 3—figure supplement 2B,C*). This is consistent with previous studies showing that the Ch bone is neural crest-derived (*Schilling and Kimmel, 1994*). However, we also detect unconverted cells in the marrow, consistent with contribution of non-neural crest-derived cells such as the mesoderm-derived vasculature (*Figure 2*).

As *sox10*:CreERT2-mediated conversion at 14 dpf also labels cells outside the cartilage, which could also contribute to osteoblasts and adipocytes, we next examined animals with lower conversion efficiency to follow discrete growth plate clones. When analyzed at 30 mm SL, growth plate clones could be quite large, consistent with clonal selection as described in the zebrafish heart and

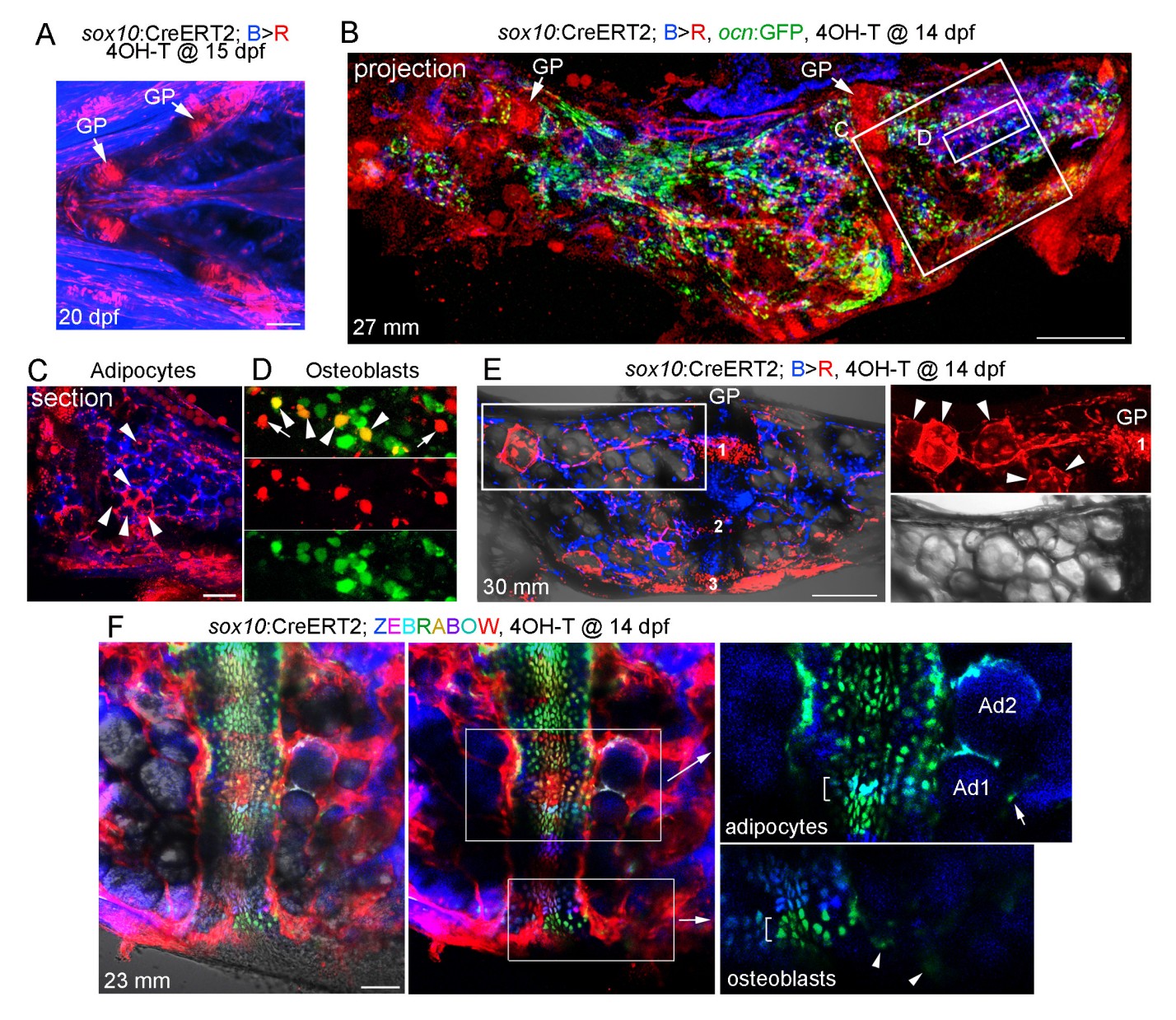

**Figure 3.** Contribution of *sox10*+ chondrocytes to osteoblasts and marrow adipocytes. (A) Confocal projection of a *sox10*:CreERT2; *bactin2*:tagBFP>DsRed animal treated at 15 dpf with 4-OHT and imaged at 20 dpf. A ventral view of the lower face shows conversion in growth plate (GP) Ch chondrocytes, as well as additional mesenchymal cells throughout the face. *n* = 3. (B) Confocal projection of a dissected Ch bone from a *sox10*:CreERT2; *bactin2*:tagBFP>DsRed; *ocn*:GFP animal converted at 14 dpf and imaged as an adult (27 mm SL). In addition to labeling of the growth plates, extensive DsRed+ cells are seen throughout the Ch in 3/3 strongly converted animals. (C) Higher magnification confocal section through the boxed region in (B) shows a subset of adipocytes labeled by DsRed (red, arrowheads). (D) Higher magnification confocal section through the boxed region in (B) shows a mixture of converted (yellow, arrowheads) and unconverted (green) *ocn*:GFP + osteoblasts, as well as converted *ocn*:GFP- mesenchymal cells (red, arrows). (E) Confocal projection of a dissected Ch bone from a *sox10*:CreERT2; *bactin2*:tagBFP >DsRed animal converted at 14 dpf and imaged as an adult (30 mm SL). Three prominent clones in the growth plate are numbered. In the boxed regions to the right, a discrete clone of growth plate chondrocytes transitions into a stream of mesenchymal cells and then a number of adipocytes (arrowheads). The brightfield image from the same sample (below) shows the lipid vesicles characteristic of adipocytes. Similar clonal contributions were seen in four independently converted animals. (F) Confocal projection of a portion of a dissected Ch growth plate from a *sox10*:CreERT2; Zebrabow animal converted at 14 dpf and imaged as an adult (23 mm SL). Images are shown with and without the Nomarski channel. Unconverted cells are red, and distinctly colored growth plate clones are visible. Magnified images corresponding to the boxed regions are shown without the red channel to highlight distinct green and teal clones (brackets). The teal clone of growth plate chondrocytes is contiguous with two similarly colored adipocytes (Ad1, Ad2), and the green clone is contiguous with faintly

*Figure 3 continued on next page*

*Figure 3 continued*

green cells (arrowheads) in cortical bone. In the adipocyte clone, the arrow indicates a green marrow cell distinct from the teal-colored adipocytes. Comparable clonal contributions were seen in three independently converted animals. Scale bars = 100 μm (A), 200 μm (B,E,F), 50 μm (C).

DOI: https://doi.org/10.7554/eLife.42736.006

The following figure supplements are available for figure 3:

**Figure supplement 1.** Characterization of the *sox10*:CreERT2 and *col2a1a*:CreERT2 transgenic lines.

DOI: https://doi.org/10.7554/eLife.42736.007

**Figure supplement 2.** Neural crest contributions to the Ch bone and marrow adipocytes.

DOI: https://doi.org/10.7554/eLife.42736.008

skeletal muscle (*Gupta and Poss, 2012*; *Nguyen et al., 2017*). In one example with three discrete clones, we observed a clone that contributed to a narrow column of growth plate chondrocytes in the middle of Ch that was contiguous with mesenchymal cells and then adipocytes toward the central marrow cavity (*Figure 3E*). To more definitively follow growth plate clones, we also examined *sox10*:CreERT2; *ubb*:Zebrabow fish in which 4OH-T treatment at 14 dpf resulted in conversion of RFP (red) to various color combinations of CFP, YFP, and RFP at 23 mm SL. Analysis of uniquely colored clones in the growth plate revealed those that contained both chondrocytes and adjacent adipocytes in the marrow (*Figure 3F*). We also observed clones that appeared to contain chondrocytes and weakly labeled cells embedded in cortical bone, though the clonal contribution to these and other mesenchymal lineages will require further analysis. Together, our data are consistent with chondrocytes giving rise to adipocytes and mesenchymal cells, and potentially osteoblasts, after growth plate remodeling in zebrafish.

## Contribution of *col2a1a+* chondrocytes to osteoblasts, adipocytes, and mesenchymal cells

As *sox10*-CreERT2-mediated conversion was broader than just the cartilage, we also generated an inducible *col2a1a*-CreERT2 line to more precisely trace chondrocytes and their derivatives. In mice, *Col2a1*:CreERT2-mediated conversion at embryonic and early postnatal stages broadly labels not only chondrocytes but also osteochondroprogenitors, such as those in the perichondrium and periosteum (*Ono et al., 2014*). In zebrafish, *col2a1a* is similarly expressed at high levels in chondrocytes and in weaker levels in osteoblasts and perichondrium, although direct evidence for *col2a1a* marking osteochondroprogenitors in zebrafish is lacking (*Eames et al., 2012*). In order to restrict expression to chondrocytes, thus avoiding potential complications of labeling osteoblasts and putative perichondral progenitors, we utilized a chondrocyte-specific 'R2' enhancer of the zebrafish *col2a1a* gene that we had previously characterized (*Dale and Topczewski, 2011*; *Askary et al., 2015*). Treatment of *col2a1a/B > R* fish with a single dose of 4-OHT at five dpf resulted in extensive labeling of *col2a1a-BAC*:GFP+ chondrocytes at 12 dpf, but no labeling of the perichondrium, periosteum, and osteoblasts as marked by the *sp7*:GFP transgene (*Figure 4A,B*; *Figure 3—figure supplement 1B*). We also observed labeling of the notochord in larval fish, but no labeling of the vasculature, blood, or other tissues examined. After conversion of *col2a1a/B > R* chondrocytes at five dpf and examination at adulthood (30 mm SL), maximal intensity projections through Ch revealed extensive labeling of growth plate chondrocytes, as well as cells throughout the bone and marrow cavity (*Figure 4C*). Thus, the majority of adult growth plate chondrocytes in Ch appear to derive from embryonic chondrocytes. Moreover, optical sections revealed cytoplasmic DsRed staining in lipid-filled adipocytes, presumptive osteoblasts lining the inner surface of bone, and mesenchymal cells within the marrow cavity. In some animals displaying lower conversion efficiency, we observed apparent clones of cells containing growth plate chondrocytes, large adipocytes, and osteoblasts embedded in Calcein +mineralized matrix (*Figure 4D*). Analysis of individual sections at higher magnification revealed contribution of *col2a1a*-lineage cells to a subset of osteoblasts within both the endosteal and periosteal surfaces of bone, as well as embedded osteocytes with characteristic cellular processes (*Figure 4E–G*).

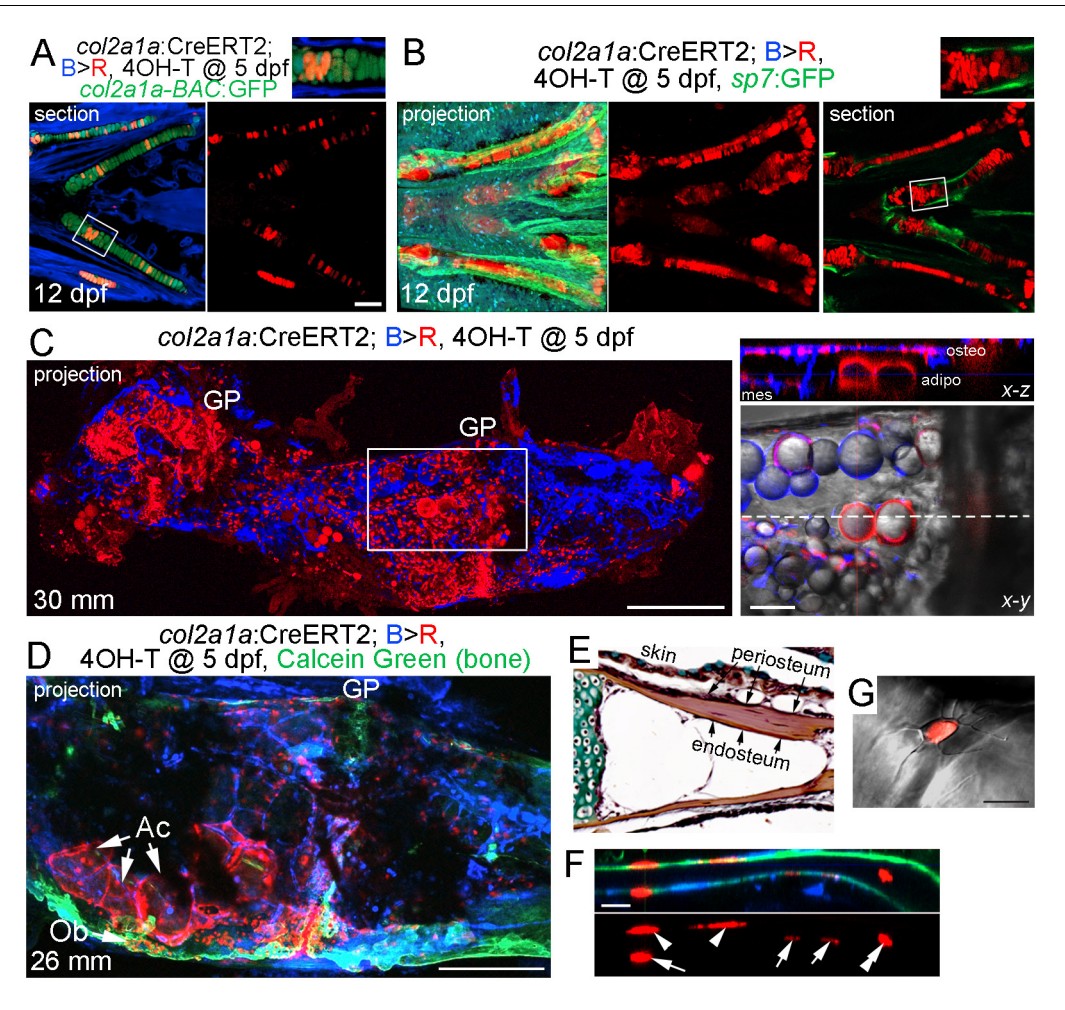

**Figure 4.** Contribution of *col2a1a*+ chondrocytes to osteoblasts and marrow adipocytes. (**A, B**) Ventral views of *col2a1a*:CreERT2; *bactin2*:tagBFP >DsRed animals treated at 5 dpf with 4-OHT and imaged at 12 dpf. Confocal sections and projections as indicated demonstrate specific conversion (red) throughout cartilage, as shown by co-localization with the chondrocyte-specific marker *col2a1a-BAC*:GFP (**A**) and lack of co-localization with the osteoblast and periosteum marker *sp7*:GFP (**B**). Boxed areas are magnified in the top right insets. *n* = 6 for each. (**C**) After conversion of *col2a1a*:CreERT2; *bactin2*:tagBFP>DsRed animals at five dpf, a confocal projection through the dissected Ch of an adult (30 mm SL) shows extensive DsRed+ cells in the growth plates (GP) and throughout the bone. A higher magnification view of the boxed region, along with brightfield, shows DsRed fluorescence in the thin cytoplasm surrounding the prominent lipid vesicles indicative of marrow adipocytes, as well as in osteoblasts (osteo) of cortical bone and mesenchymal cells (mes) within the marrow cavity. The dashed line in the *x-y* slice shows the position of the *x-z* slice above. *n* = 10. (**D**) In this example of a *col2a1a*:CreERT2; *bactin2*: tagBFP>DsRed animal converted at five dpf and imaged as an adult, a prominent clone of DsRed+ cells are evident at the bottom of the growth plate, consisting of GP chondrocytes, adipocytes (Ac), and osteoblasts (Ob) associated with Calcein Green+ cortical bone. Similar clonal contributions were seen in four independently converted animals. (**E**) Pentachrome staining of a portion of the Ch growth plate at 19 mm SL shows the endosteum and periosteum. Note that zebrafish have osteocytes embedded in their cortical bone (the dark nuclei in the reddish-brown matrix). (**F, G**) High-magnification images of a section of Ch cortical bone from an animal converted at five dpf and imaged at 28 mm SL. DsRed+ osteoblasts/osteocytes are seen in the endosteal surface (arrows), periosteal surface (arrowheads), and embedded in bone (double arrowhead). The merged brightfield and fluorescence image from a different example (**G**) shows a DsRed+ cell with cellular processes characteristic of osteocytes. Scale bars = 100 μm (**A,B**), 200 μm (**C,D**), 50 μm (**F**), 20 μm (**G**).
DOI: https://doi.org/10.7554/eLife.42736.009

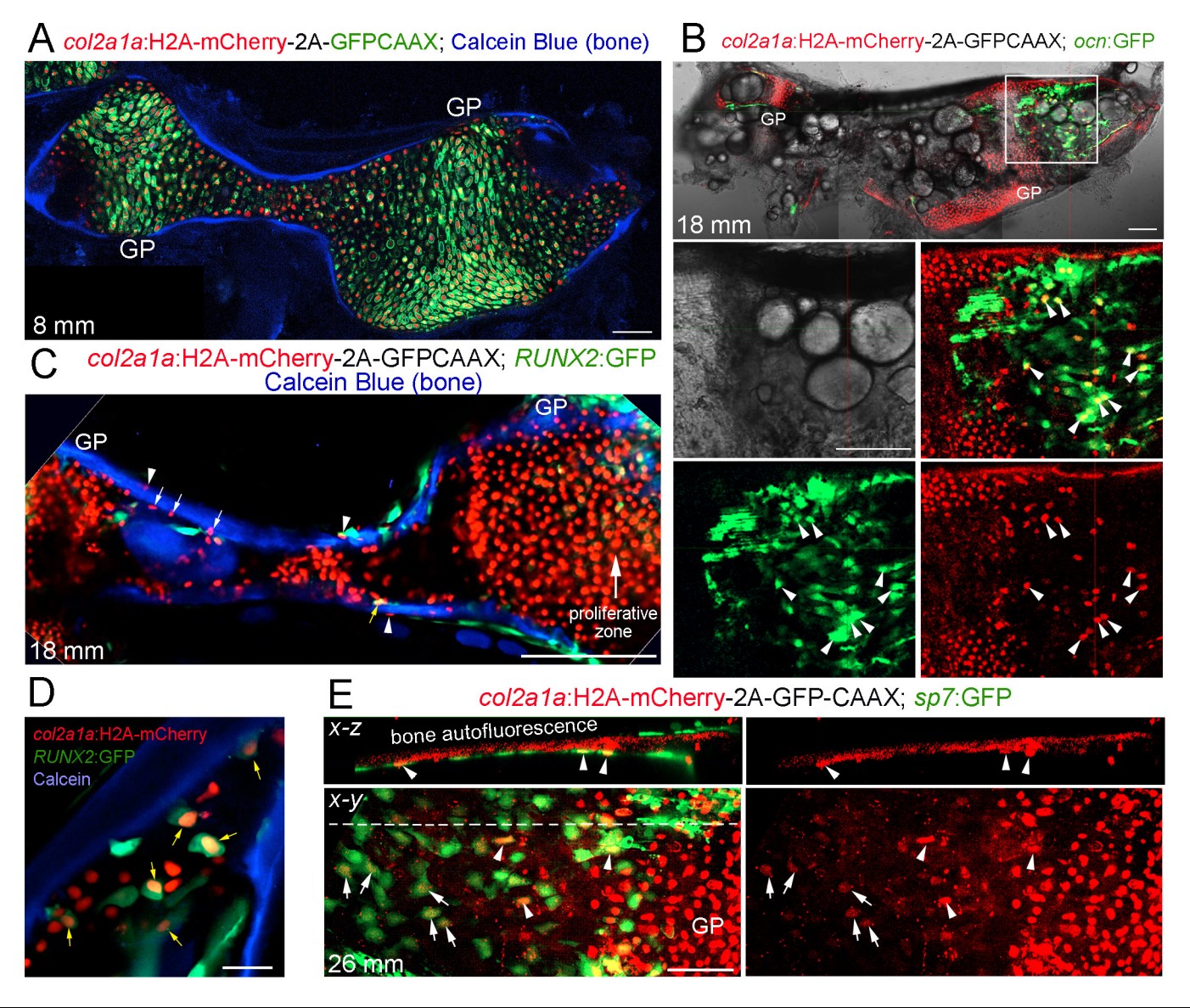

**Figure 5.** Tracing of *col2a1a*-lineage cells by a long-lived Histone2A-mCherry fusion protein. (**A**) At a stage preceding growth plate remodeling (8 mm SL), *col2a1a*:H2A-mCherry-2A-GFPCAAX labels chondrocytes, but not osteoblasts associated with Calcein Blue+ mineralized bone.rowth plate (GP) chondrocytes co-express the nuclear Histone2A-mCherry protein (red) and the membrane-localized GFPCAAX protein (green). In the middle and poles of Ch, hypertrophic chondrocytes retain the long-lived H2A-mCherry protein but not the short-lived GFPCAAX protein, reflective of the down-regulation of *col2a1a* expression during hypertrophic maturation. *n* = 3. (**B**) In a confocal section through the dissected Ch of a juvenile fish (18 mm SL), numerous H2A-mCherry+; *ocn*:GFP+ cells are seen in regions where the cartilage template is being converted to fat. Magnification of the boxed region shows the brightfield image (white), a merged image of H2A-mCherry+ cells (red) and *ocn*:GFP+ cells (green), and individual channels below. We observed a number of H2A-mCherry+; *ocn*:GFP+ cells (arrowheads) in 4/4 animals. (**C**) Confocal projection of a dissected Ch at 18 mm SL reveals cells expressing nuclear Histone2A-mCherry (red) on both the endosteal surface (arrows) and periosteal surface (arrowheads) of Calcein Blue+ cortical bone. Some H2A-mCherry+ cells associated with bone also co-express the osteoprogenitor marker *RUNX2*:GFP (yellow arrow). Note that the membrane GFPCAAX signal from the *col2a1a*:H2A-mCherry-2A-GFPCAAX transgene is much weaker and barely detectable in the proliferative zone at the gain settings used to image cytoplasmic *RUNX2*:GFP. *n* = 3. (**D**) Confocal section through the Ch at higher magnification shows several H2A-mCherry+; *RUNX2*:GFP+ cells (yellow arrows) in the marrow cavity and close to the endosteal surface of the Calcein Blue+ bone. (**E**) In adult fish (26 mm SL), several H2A-mCherry+ cells are found to co-express the osteoblast marker *sp7*:GFP on the endosteal surface. H2A-mCherry tends to be stronger closer to the growth plate; arrowheads denote stronger and arrows denote weaker H2A-mCherry signal. The white dotted line in the *x-y* section shows the location of the *x-z* section above. *n* = 5. Scale bars = 50 μm (**A,D,E**), 100 μm (**B,C**).

DOI: https://doi.org/10.7554/eLife.42736.010

The following figure supplement is available for figure 5:

*Figure 5 continued on next page*

*Figure 5 continued*

**Figure supplement 1.** Characterization of the *col2a1a*:Histone2A-mCherry-2A-GFPCAAX line.

DOI: https://doi.org/10.7554/eLife.42736.011

## Long-lived Histone2A-mCherry protein reveals contribution of *col2a1a* + chondrocytes to osteoblasts

A limitation of CreER-mediated lineage tracing is that it can be difficult to rule out contributions from rare converted cells outside the population of interest, for example in the perichondrium and periosteum. We therefore employed a Cre-independent approach to independently assess the fate of growth plate chondrocytes. To do so, we expressed a Histone2A-mCherry fusion protein in *col2a1a*+ cells. An advantage of this type of lineage approach is that the levels of Histone2A-mCherry protein, which is stably incorporated into chromatin, reflect those of endogenous *col2a1a* in chondrocytes provided continued cell division does not dilute out the fusion protein. This is in contrast to Cre-mediated approaches in which a strong ubiquitous promoter determines the level of a reporter protein; hence, even low levels of Cre recombinase activity outside of chondrocytes (e.g. in osteochondroprogenitors) can result in strong reporter expression. In zebrafish, *col2a1a* is expressed at high levels in the proliferative zone of the Ch growth plate and then downregulated in the hypertrophic zone (*Paul et al., 2016*). In a newly generated *col2a1a*:Histone2A-mCherry-T2A-GFP-CAAX transgenic line, membrane-localized GFP-CAAX, which is rapidly turned over, is seen primarily in the proliferative zone, whereas Histone2A-mCherry is seen uniformly throughout the Ch cartilage, confirming the long-lived nature of this fusion protein (*Figure 5A* and *Figure 5—figure supplement 1A,B*). At 7–8 mm SL (approx. three wpf), which is well before the start of growth plate remodeling at 11–12 mm SL, all Ch chondrocytes are Histone2A-mCherry+ and we do not detect Histone2A-mCherry+ cells associated with Calcein Blue+ bone or co-expressing the osteoblast transgene *ocn*:GFP (*Figure 5A* and *Figure 5—figure supplement 1C,D*). In contrast, at post-remodeling stages (12 and 18 mm SL), we observe extensive overlap of Histone2A-mCherry with *ocn*:GFP + osteoblasts associated with cortical bone (*Figure 5B* and *Figure 5—figure supplement 1E*). We also observe numerous Histone2A-mCherry+ cells embedded in the endosteal and periosteal surfaces of the Ch bone (labeled by Calcein Blue), with several of these cells co-expressing the pre-osteoblast transgene *RUNX2*:GFP or the early osteoblast transgene *sp7*:GFP (*Figure 5C–E*). Note that *ocn*:GFP, *RUNX2*:GFP, and *sp7*:GFP can all be readily distinguished from membrane GFP-CAAX by their much stronger and cytoplasmic expression. In addition, we observed that *sp7*:GFP+ osteoblasts further from the growth plate tended to have weaker Histone2A-mCherry signal than those more closely associated with the edge of the hypertrophic zone, suggesting that hypertrophic chondrocytes and/or their osteoblast derivatives undergo cell division to dilute out the Histone2A-mCherry signal. These findings independently confirm the conclusions of our CreER lineage tracing studies that *col2a1a*+ chondrocytes generate osteoblasts in zebrafish.

## Hypertrophic chondrocytes re-enter the cell cycle and express *lepr*

The conversion of hypertrophic chondrocytes to osteoblasts and adipocytes could occur in the absence of cell division (i.e. 'transdifferentiation') and/or through partial dedifferentiation into a proliferative progenitor. To test these possibilities, we first used incorporation of bromodeoxyuridine (BrdU) to detect proliferative cells in the Ch during remodeling stages (*Figure 6A,B*). At the beginning of remodeling (11 mm SL), we detected BrdU+ cells in the central zone of chondroblasts in the growth plate, as well as in the perichondrium and periosteum. In addition, we observed BrdU+ cells at the edge of the hypertrophic zone where the cartilage matrix is being actively degraded, similar to what has been reported in mouse (*Park et al., 2015*). We observed BrdU incorporation in similarly positioned hypertrophic chondrocytes at 15 and 19 mm SL, with BrdU+ cells becoming fewer in the perichondrium and periosteum by 19 mm.

We next tested whether hypertrophic chondrocytes that re-enter the cell cycle also express known skeletal stem cell markers. In mice, *LepR* expression marks a heterogeneous population of cells in endochondral bone, including a putative postnatal skeletal stem cell population (*Zhou et al.,*

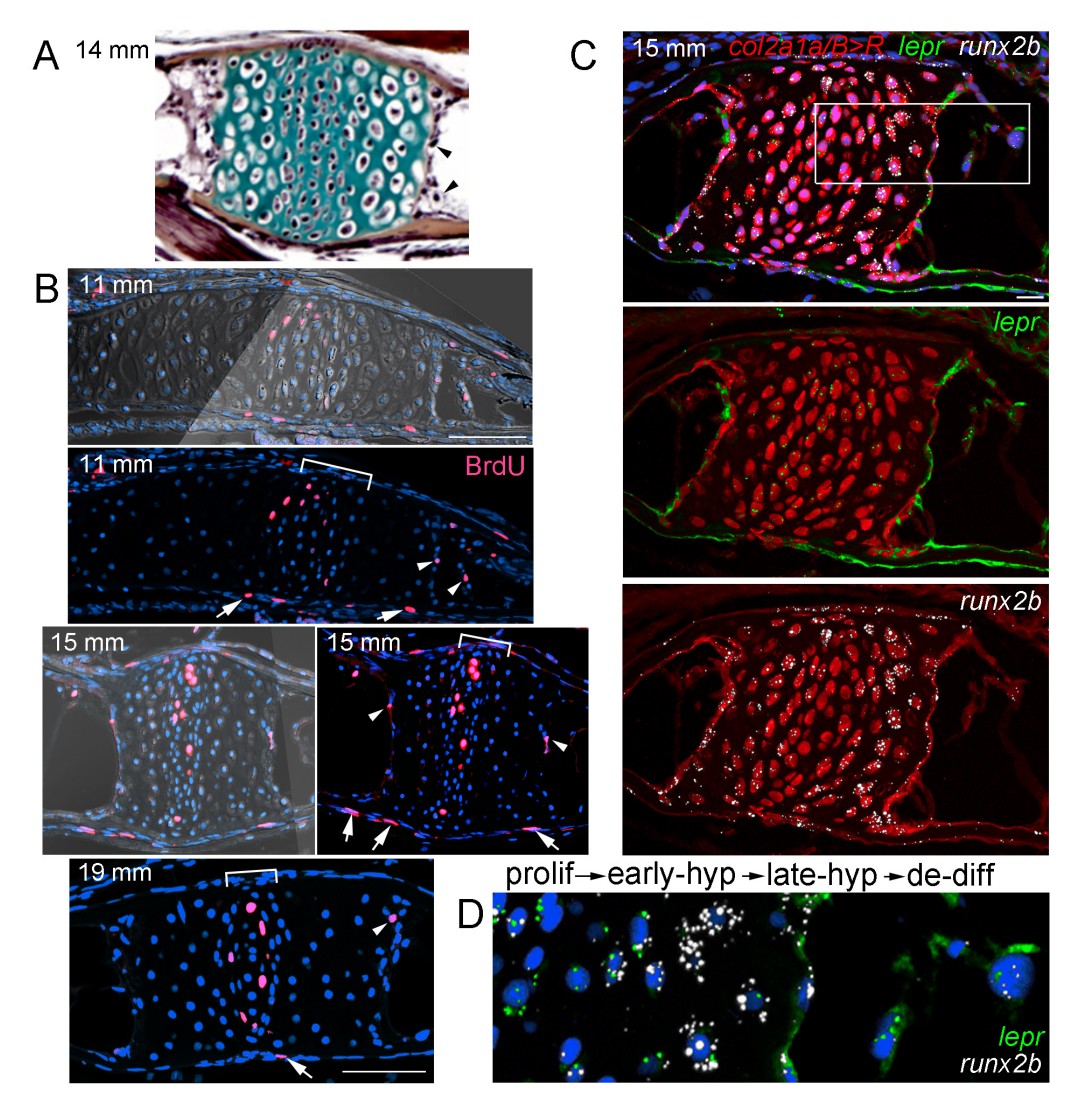

**Figure 6.** Late-stage hypertrophic chondrocytes re-enter the cell cycle and express *lepr*. (A) Pentachrome staining of a section through a Ch growth plate at 14 mm SL. Arrowheads denote two examples of hypertrophic chondrocytes at the edges of the growth plates that lack collagen-rich matrix (green) and appear to be exiting their lacunae. (B) BrdU incorporation (pink) relative to all nuclei (Hoechst, blue) shows recently divided cells. Fluorescent images with or without brightfield are shown for 11 and 15 mm SL stages, and fluorescent channel only for 19 mm SL. In addition to BrdU +cells in the proliferative zones of the growth plates (brackets) and perichondrium (arrows), a subset of hypertrophic chondrocytes at the edges of the growth plates (arrowheads) are BrdU+ at each stage. Proliferative hypertrophic chondrocytes were seen in sections from three independent animals at each stage. (C–D) Fluorescent RNAscope in situ hybridization for *lepr* (green) and the hypertrophic chondrocyte and osteoblast precursor marker *runx2b* (white). Red signal indicates cells derived from *col2a1a/B > R* chondrocytes that were converted by addition of 4-OHT at five dpf (detected by anti-DsRed antibody), and all nuclei are shown in blue (Hoechst). In a section of a Ch growth plate at 15 mm SL, the merged channel above and red/green and red/white channels below show expression of *lepr* and *runx2b* in chondrocytes and their derivatives. In the higher magnification view of the boxed region (D), *lepr* is expressed in proliferative chondrocytes, *runx2b* is expressed at high levels and *lepr* at lower levels in early hypertrophic chondrocytes, and *lepr* and *runx2b* are co-expressed in late hypertrophic chondrocytes and adjacent mesenchymal cells that have been released from the growth plate. Similar expression of *lepr* and *runx2b* was seen in sections from 4/4 independent animals. Scale bars = 50 µm (B,C), 20 µm (D).
DOI: https://doi.org/10.7554/eLife.42736.012

The following figure supplement is available for figure 6:

**Figure supplement 1.** Expression of *Lepr/lepr* mRNA in zebrafish and mouse endochondral bone.
DOI: https://doi.org/10.7554/eLife.42736.013

*2014b*). First, we examined expression of *lepr* and found dynamic expression in the zebrafish Ch endochondral bone from juvenile through adult stages. A comparison with the hypertrophic chondrocyte and early osteoblast marker *runx2b* shows higher *lepr* expression in proliferative versus hypertrophic chondrocytes at 8, 12, and 20 mm SL stages, with chondrocyte expression decreasing by 27 mm SL (*Figure 6—figure supplement 1A*). During remodeling of zebrafish Ch (15 mm SL), we also observe *lepr+* cells in the marrow cavity that are derived from chondrocytes, based on labeling by *col2a1a/B > R* (*Figure 6C*), and we continue to observe *lepr+* cells in the Ch marrow at later stages (20 and 27 mm SL) (*Figure 6—figure supplement 1A*). These results are consistent with *lepr* + cells in the bone marrow deriving from growth plate chondrocytes in zebrafish, although direct evidence will be needed to determine if any of these chondrocyte-derived *lepr+* marrow cells behave as skeletal stem cells in zebrafish.

## Requirement of *mmp9* for growth plate remodeling and marrow adipocyte formation

In mice, *Mmp9* has been reported to function in hematopoietic lineage cells for growth plate remodeling, as bone marrow transplants can rescue the delay in growth plate remodeling seen in *Mmp9* mutants (*Vu et al., 1998*). Here, we tested whether *mmp9* might have a conserved requirement for growth plate remodeling in zebrafish, including the generation of marrow adipocytes. At the beginning of remodeling (11 mm SL), we observe expression of *mmp9* at the edge of the hypertrophic zone, with this restricted expression in late-stage hypertrophic chondrocytes continuing through 17 mm SL stages (*Figure 7A*). Higher magnification views show that *mmp9* expression is prominent in hypertrophic chondrocytes that appear to be exiting their lacunae. Next, we used CRISPR/Cas9 mutagenesis to create an early frame-shift mutation in the *mmp9* gene that is predicted to abolish most if not all protein function (*Figure 7B*). *mmp9* homozygous mutants are adult viable and do not display obvious larval craniofacial defects. Whereas trichrome staining revealed no significant differences in the mutant Ch growth plates at 17 mm SL, by 21 mm we observed that the hypertrophic zone was significantly larger, compared to the proliferative zone, in *mmp9* mutants versus controls, indicating a delay in growth plate remodeling (*Figure 7C,D*). The defect in growth plate remodeling was still evident at 27 mm SL, with mutants displaying a wider Ch growth plate (*Figure 7E,G*). Strikingly, *mmp9* mutants also had fewer adipocytes in the central marrow cavity compared to stage-matched controls (*Figure 7E,G*), consistent with adipocytes deriving from hypertrophic chondrocytes that are released from the cartilage matrix by Mmp9 activity.

Given *mmp9* expression in late-stage hypertrophic chondrocytes, we next tested whether Mmp9 might function in chondrocytes as opposed to hematopoietic lineage cells for growth plate remodeling and marrow adipocyte generation in zebrafish. As the Ch bone is generated from neural crest-derived cells, we used transplantation of ubiquitously labeled wild-type ectodermal cells into the neural crest precursor domain of unlabeled *mmp9-/-* shield-stage hosts to generate wild-type Ch bones in otherwise mutant hosts. At adult stages, we were able to recover eight mutant recipients with contribution of wild-type cells to chondrocytes of the Ch growth plate. In these animals, we observed a rescue of growth plate width, with a trend toward better rescue in wild-type versus mutant regions of the growth plate (p = 0.06), as well as a trend toward rescue of adipocyte number (p = 0.06) (*Figure 7E–G*). As a control, transplantation of wild-type neural crest cells into wild-type animals had no effect on Ch growth plate width and adipocyte number. These results indicate that *mmp9* is required in the neural crest lineage, and potentially chondrocytes themselves, for efficient remodeling of the growth plate and the generation of marrow adipocytes from chondrocytes.

## Discussion

Despite anatomical differences between zebrafish and mammalian bones, we find that growth plate remodeling is remarkably well conserved and thus likely ancestral to bony vertebrates. The zebrafish Ch undergoes a transient breakdown of cortical bone near the growth plates, which coincides with extensive vascularization and an Mmp9-dependent replacement of hypertrophic chondrocytes with fat and bone. Using multiple methods of lineage tracing, including a Cre-independent technique, we show that late-stage hypertrophic chondrocytes generate not only osteoblasts but also marrow adipocytes in zebrafish. Further support for the ability of chondrocytes to generate adipocytes is that delayed growth plate remodeling in *mmp9* mutants results in a paucity of marrow adipocytes in

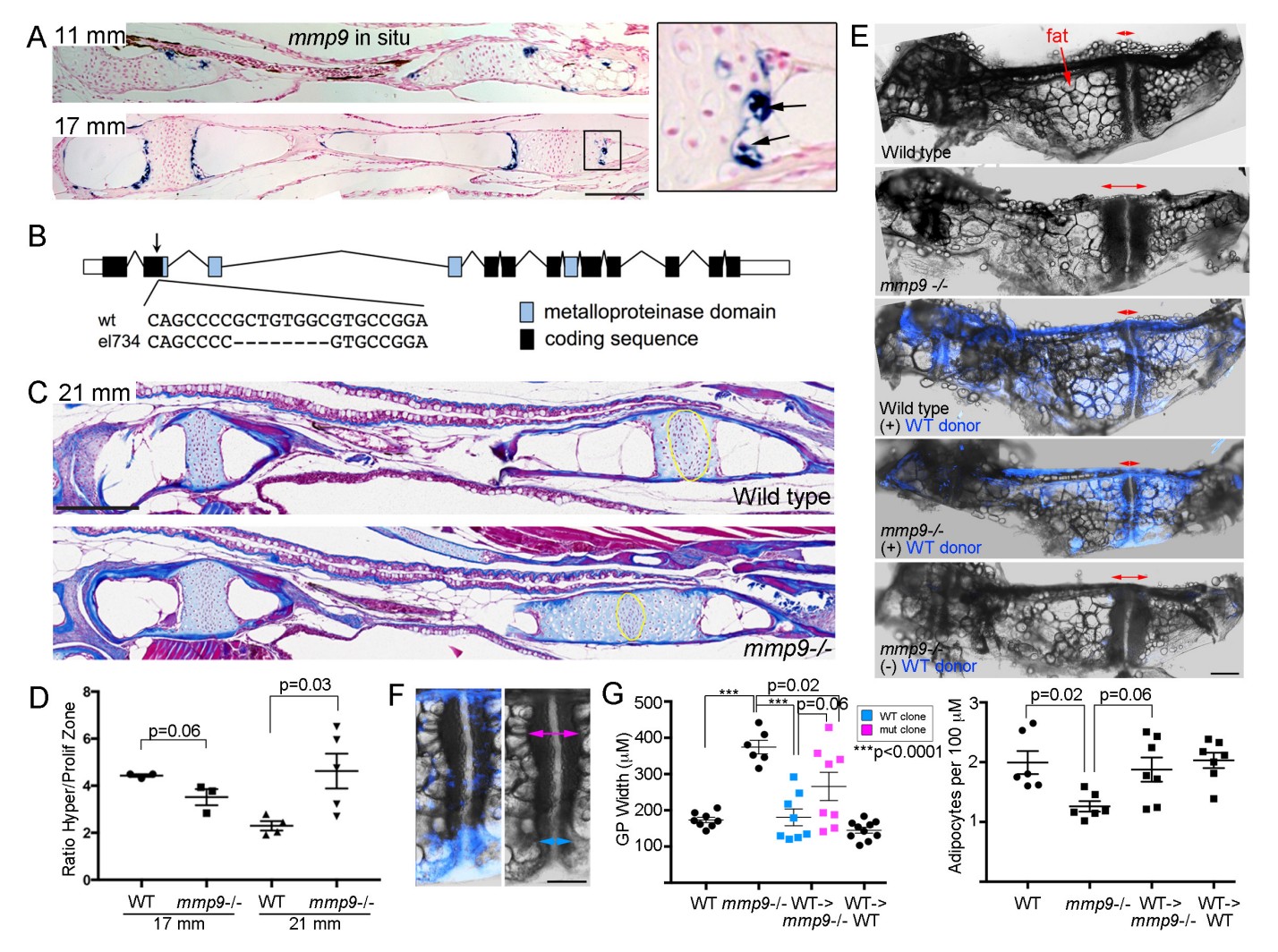

**Figure 7.** Tissue-autonomous requirement for *mmp9* in cartilage remodeling. (A) Colorimetric mRNA in situ hybridization shows expression of *mmp9* (blue) in sections of the Ch at 11 and 17 mm SL. Inset shows specific expression of *mmp9* in hypertrophic chondrocytes (arrows) at the edge of the growth plate. Nuclear fast red was used as a counterstain. n = 2 at each stage. (B) Schematic of the *mmp9* gene locus in zebrafish. Rectangles denote exons. The site of the 8 bp deletion in the *el734* allele is indicated by an arrow, with specific sequence changes shown below. This frame-shift mutation is predicted to result in an early stop codon and loss of the catalytic metalloproteinase domain. (C) Trichrome staining at 21 mm SL shows enlarged growth plates in the Ch bones of *mmp9* mutants. The approximate regions of the proliferative zones used for quantification in (D) are shown by the yellow ovals. (D) Quantification of the ratio of the hypertrophic to the proliferative zones shows a delay in remodeling the hypertrophic zone in *mmp9* mutants at 21 but not 17 mm SL. We performed a students t-test and show standard error of the mean. (E,F) Dissected Ch bones at adult stages (27–31 mm SL) from wild-type, *mmp9* mutant, and wild-type and mutant hosts receiving wild-type donor neural crest transplants (blue). Ectoderm cells from *bactin2*:tagBFP >DsRed donors were transplanted unilaterally into the neural crest precursor domain of unlabeled hosts at six hpf. Red two-sided arrows indicate the width of the posterior growth plates. (+) denotes sides receiving transplants, and (-) denotes contralateral control sides. The rescued growth plate from the *mmp9* mutant receiving a wild-type neural crest transplant is shown at higher magnification in (F), with blue arrows showing a narrower wild-type growth plate clone and magenta arrows a wider mutant clone. (G) Quantification shows that *mmp9* mutants have wider growth plates and fewer adipocytes in the marrow than wild-type siblings. Wild-type neural crest transplants rescue growth plate width in *mmp9* mutants, with a trend toward better rescue in areas of the growth plate with wild-type (blue) versus mutant (magenta) clones. There was also a strong trend toward rescue of adipocyte number with wild-type neural crest transplants that contributed to growth plate chondrocytes. We performed a Tukey-Kramer HSD test and show standard error of the mean. Unless indicated, all other comparisons were not significant (p > 0.05). Scale bars = 50 μm (**A,C**), 200 μm (**E, F**). See also *Figure 7—source data 1*.

DOI: https://doi.org/10.7554/eLife.42736.014

The following source data is available for figure 7:

**Source data 1.** Quantification of growth plate width and adipocyte numbers in *mmp9* mutants and rescued experiments.
DOI: https://doi.org/10.7554/eLife.42736.015

adults. Lastly, we show that hypertrophic chondrocytes re-enter the cell cycle and contribute to *lepr* + mesenchymal cells, raising the possibility that partial dedifferentiation into proliferative progenitors underlies chondrocyte fate transitions inside endochondral bones.

As zebrafish have hollow bones that lack hematopoiesis, one possibility was that most if not all of the cortical bone of adult zebrafish would be simply derived from the periosteum. However, we find significant contribution of chondrocyte-derived cells to both the endosteal and periosteal surfaces of the Ch bone, similar to what has been described in the mouse (*Yang et al., 2014*; *Zhou et al., 2014a*; *Jing et al., 2015*). We also reveal chondrocytes to be a significant source of marrow adipocytes in zebrafish. Although the incomplete conversion efficiency of the CreER lines, as well as the expression of *sox10*:CreER outside of chondrocytes, made it difficult to precisely quantify what proportion of osteoblasts and marrow adipocytes derive from chondrocytes, there are likely other sources for these cells, in particular the periosteum which houses several types of skeletal stem cells (*Debnath et al., 2018*). Our findings raise the question of whether marrow adipocytes also derive in part from chondrocytes in mammals. Indeed, older studies have demonstrated the ability of murine chondrocytes to differentiate into adipocytes in vitro (*Heermeier et al., 1994*; *Hegert et al., 2002*). Further, *Col2a1*:CreER cells converted at postnatal day three in mouse were found to give rise to adipocytes in the metaphyseal bone marrow, although a caveat is that *Col2a1*:CreER marks both chondrocytes and progenitors at early postnatal stages (*Ono et al., 2014*).

The finding that late-stage hypertrophic chondrocytes can re-enter the cell cycle and express *lepr* suggests that at least some of these cells may dedifferentiate into stem-like cells, which subsequently expand in number and differentiate into osteoblasts and adipocytes. Future molecular profiling of these chondrocyte-derived marrow mesenchymal cells will be needed to better characterize their relationship to previously identified skeletal stem cells. We also cannot rule out that some hypertrophic chondrocytes directly change into adipocytes and/or osteoblasts in the absence of cell division (i.e. 'transdifferentiation'). Indeed, the detection of osteoblasts with strong *col2a1a*:Histone2A-mCherry signal suggests that in some cases chondrocytes can form osteoblasts with little to no cell division, as otherwise the Histone2A-mCherry signal would have been diluted out with successive cell divisions. On the other hand, osteoblasts farther from the growth plate tended to have weaker Histone2A-mCherry signal, suggesting proliferation of a progenitor intermediate, and we directly observed late-stage hypertrophic chondrocytes undergoing cell division. Whereas it has been suggested in mouse that only chondrocytes near the periosteal surface, that is 'borderline chondrocytes', may be capable of lineage plasticity (*Bianco et al., 1998*; *Maes et al., 2010*), we detected lineage clones containing growth plate chondrocytes, mesenchymal cells, and adipocytes in the central part of the growth plate (*Figure 3E,F*), arguing against such a model.

A notable feature of *LepR*-lineage cells in mice is that they contribute to osteoblasts and adipocytes primarily in postnatal phases, that is when growth plate remodeling is already underway (*Yang et al., 2014*). It would therefore be interesting to test whether *LepR*-expressing skeletal stem cells have a similar origin from hypertrophic chondrocytes in mammals. One caveat is that we detect endogenous *lepr* expression in both marrow cells and growth plate chondrocytes in zebrafish. However, the more specific labeling of marrow cells by the *LepR*-Cre in mouse likely reflects the Cre insertion being in the long *LepR* isoform (containing exon 18b) that displays more restricted expression than the short isoform (*Zhou et al., 2014b*). Indeed, *LepR* mRNA and protein has also been reported in chondrocytes of mouse (*Hoggard et al., 1997*), rat, and human (*Morroni et al., 2004*), a finding we confirmed in the postnatal mouse femur, including the same higher expression in immature versus hypertrophic chondrocytes that we observe in zebrafish (*Figure 6—figure supplement 1B*). Without a comparable long-isoform *lepr*-Cre line in zebrafish, we cannot therefore conclude whether zebrafish *lepr*+ marrow cells are comparable to those described in mouse. The future generation of Cre lines to specifically mark *lepr*+ marrow cells in zebrafish will be needed to determine whether these chondrocyte-derived marrow cells also act as stem cells for osteoblasts and adipocytes in post-embryonic fish.

A similar delay in the remodeling of the hypertrophic zone in mouse *Mmp9* and zebrafish *mmp9* mutants reveals genetic conservation of growth plate remodeling from fish to mammals. In contrast to mice, we find that Mmp9 in zebrafish appears to function primarily in chondrocytes for growth plate remodeling. We also observed rescue of marrow adipocyte number by wild-type chondrocytes, though this had moderate statistical significance (p = 0.06), potentially owing to low sample size (*n* = 8), the mosaic contribution of wild-type cells to the growth plate, and/or roles of cells outside

the growth plate to marrow adipocyte generation. In mice, loss of *Mmp13* in chondrocytes does result in a growth plate remodeling delay, with global *Mmp13* deletion enhancing the remodeling defects of *Mmp9* mutants (*Inada et al., 2004*; *Stickens et al., 2004*). It may be that MMPs are derived from both chondrocytes and invading hematopoietic cells (e.g. osteoclasts), with the relative importance of each cell source varying between zebrafish and mammals. Compensation by *mmp13* might also explain why we detected growth remodeling defects in *mmp9* zebrafish mutants at 21 and 27–31 mm SL stages, but not at 17 mm SL. Mmp9 and Mmp13 have known roles in degrading components of the cartilage extracellular matrix (*Page-McCaw et al., 2007*), consistent with our observed loss of collagen-rich matrix in hypertrophic chondrocytes at the edges of the zebrafish growth plate. Secreted Mmp9 may therefore function simply to degrade cartilage matrix and facilitate release of dedifferentiating chondrocytes into the marrow. Intriguingly, Mmp9 has recently been shown to have an additional, non-canonical function in the nucleus for histone H3 tail cleavage. Whereas this has only been demonstrated so far for osteoclasts (*Kim et al., 2016*), it remains possible that Mmp9 could have a similar non-canonical function in altering the chromatin structure of hypertrophic chondrocytes to allow them to acquire new potential. In conclusion, our data support conservation of mammalian-like growth plate remodeling in zebrafish, which provides new opportunities for better understanding the molecular and cellular mechanisms by which hypertrophic chondrocytes transform into osteoblasts, marrow adipocytes, and potentially adult skeletal stem cells within endochondral bones.

# Materials and methods

**Key resources table**

| Reagent type (species) or resource | Designation | Source or reference | Identifiers | Additional information |
|---|---|---|---|---|
| Genetic reagent (*D. rerio*) | *sp7:EGFP* | PMID: 20506187 | RRID: ZFIN ID: ZDB-GENO-100402–2 | Zebrafish International Resource Center |
| Genetic reagent (*D. rerio*) | *Hsa.RUNX2:EGFP* | PMID: 23155370 | RRID: ZFIN ID: ZDB-ALT-120209–60 | Zebrafish International Resource Center |
| Genetic reagent (*D. rerio*) | *Mmu.Sox10-Mmu.Fos:Cre* | PMID: 23155370 | RRID: ZFIN ID: ZDB-ALT-130614–2 | Zebrafish International Resource Center |
| Genetic reagent (*D. rerio*) | *Ola.Osteocalcin.1:EGFP* | PMID: 21571227 | RRID: ZFIN ID: ZDB-ALT-110713–1 | Zebrafish International Resource Center |
| Genetic reagent (*D. rerio*) | *col2a1a BAC:GFP* | PMID: 26555055 | RRID: ZFIN ID: ZDB-ALT-160204–6 | Zebrafish International Resource Center |
| Genetic reagent (*D. rerio*) | *col2a1aBAC: mCherry-NTR* | PMID: 26555055 | RRID: ZFIN ID: ZDB-ALT-160204–7 | Zebrafish International Resource Center |
| Genetic reagent (*D. rerio*) | *bactin2:loxP-BFP-loxP-DsRed* | PMID: 25119047 | RRID: ZFIN ID: ZDB-ALT-141111–8 | Zebrafish International Resource Center |
| Genetic reagent (*D. rerio*) | *fli1a:eGFP* | PMID: 12167406 | RRID: ZFIN ID: ZDB-ALT-060810–2 | Zebrafish International Resource Center |

*Continued on next page*

*Continued*

| Reagent type (species) or resource | Designation | Source or reference | Identifiers | Additional information |
|---|---|---|---|---|
| Genetic reagent (*D. rerio*) | *kdrl:eGFP* | PMID: 16251212 | RRID: ZFIN ID: ZDB-ALT-061120–6 | Zebrafish International Resource Center |
| Genetic reagent (*D. rerio*) | *(−5.2)lyve 1b:DsRed* | PMID: 22627281 | RRID: ZFIN ID: ZDB-ALT-120723–3 | Zebrafish International Resource Center |
| Genetic reagent (*D. rerio*) | *ubb:LOX2272-LOXP-RFP-LOX2272-CFP-LOXP-YFP* | PMID: 23757414 | RRID: ZFIN ID: ZDB-ALT-130816–2 | Zebrafish International Resource Center |
| Genetic reagent (*D. rerio*) | *sox10:CreERT2: bactin2:loxP-BFP-loxP-DsRed* | this paper | | allele *el777* |
| Genetic reagent (*D. rerio*) | *col2a1a-R2-E1b: CreERT2::bactin 2:loxP-BFP-loxP-DsRed* | this paper | | allele *el691* |
| Genetic reagent (*D. rerio*) | *col2a1a-R2-E1b:CreERT2:: bactin2:loxP-BFP-loxP-DsRed* | this paper | | allele *el713* |
| Genetic reagent (*D. rerio*) | *col2a1a-R2-E1b: CreERT2::bactin2: loxP-BFP-loxP-DsRed* | this paper | | allele *el712* |
| Genetic reagent (*D. rerio*) | *col2a1a-R2-E1b: H2A.F/Z-mCherry-P2A-GFPCAAX* | this paper | | allele *el690* |
| Genetic reagent (*D. rerio*) | *col2a1a-R2-E1b: H2A.F/Z-mCherry-P2A-GFPCAAX* | this paper | | allele *el695* |
| Genetic reagent (*D. rerio*) | *mmp9$^{-/-}$* | this paper | | allele *el734*; gRNA target 5'-TTGATGCCATGA AGCAGCCC-3' |
| Recombinant DNA reagent | p5E-sox10 | PMID: 22589745 | RRID: ZFIN ID: ZDB-ALT-120523–6 | Zebrafish International Resource Center |
| Recombinant DNA reagent | pDestTol2-col2a1aR2-E1B-eGFPpA | PMID: 21723274 | RRID: ZFIN ID: ZDB-ALT-111205–4 | Zebrafish International Resource Center |
| Antibody | rat anti-BrdU | Bio-Rad Laboratories | cat.#: MCA2060 GA; RRID: AB_10545551 | (1:100–150) |
| Antibody | rabbit anti-mCherry | Novus Biologicals | cat.#: NBP2-25157 | (1:250) |
| Antibody | goat anti-rabbit Alexa Fluor 568 | Thermo Fisher Scientific | cat.#: A-11011; RRID: AB_143157 | (1:200–500) |
| Antibody | goat anti-rat Alexa Fluor 633 | Thermo Fisher Scientific | cat.#: A21094; RRID: AB_2535749 | (1:500) |

*Continued on next page*

*Continued*

| Reagent type (species) or resource | Designation | Source or reference | Identifiers | Additional information |
|---|---|---|---|---|
| Sequence-based reagent | RNAscope Probe - Mm-Lepr | Advanced Cell Diagnostics | cat.#: 402731 | |
| Sequence-based reagent | RNAscope Probe - Dr-lepr | Advanced Cell Diagnostics | cat.#: 535311 | |
| Sequence-based reagent | RNAscope Probe - Dr-runx2b-C2 | Advanced Cell Diagnostics | cat.#: 409531-C2 | |
| Commercial assay or kit | Gomori One-Step, Aniline Blue, trichrome stain kit | Newcomer Supply | cat.#: 9176A | |
| Commercial assay or kit | Movat-Russell modified penta-chrome stain kit | Newcomer Supply | cat. #: 9150A | |
| Commercial assay or kit | RNAscope Multiplex Fluorescent Kit v2 | Advanced Cell Diagnostics | cat.#: 323110 | |
| Chemical compound, drug | Alizarin Red S | Amresco | cat.#: 9436–25G | live staining: 1 mg / 30 mL |
| Chemical compound, drug | Calcein Green | Thermo Fisher Scientific | cat.#: C481 | live staining: 1 mg / 10 mL |
| Chemical compound, drug | Calcein Blue, AM | Thermo Fisher Scientific | cat.#: C1429 | live staining: 5 mg / 10 mL |
| Chemical compound, drug | (Z)—4-Hydroxy-tamoxifen (4-OHT) | Sigma-Aldrich | cat.#: H7904 | Cre-lox Recombination: 5 uM |
| Chemical compound, drug | HCS LipidTOX Deep Red | Life Technologies | cat.#: H34477 | (1:200) |
| Chemical compound, drug | BrdU —5-Bromo-2′-deoxyuridine | Sigma-Aldrich | cat.#: B5002 | live staining: 4.5 mg / mL |
| Software | Prism | GraphPad | | |
| Software | FIJI | Image J | | |

## Zebrafish transgenic lines and *mmp9* mutants

All procedures were approved by the University of Southern California Institutional Animal Care and Use Committee. Published *Danio rerio* lines include *Tg(Has.RUNX2:EGFP)[zf259]* and *Tg(Mmu.Sox10-Mmu.Fos:Cre)[zf384]* (**Kague et al., 2012**), *Tg(sp7:EGFP)[b1212]* (**DeLaurier et al., 2010**), *Tg(Ola.Osteocalcin.1:EGFP)[hu4008]* (**Knopf et al., 2011**), *Tg(col2a1aBAC:GFP)[el483]* and *Tg(col2a1aBAC:mCherry-NTR)[el559]* (**Askary et al., 2015**), *Tg(bactin2:loxP-BFP-loxP-DsRed)[sd27]* (**Kobayashi et al., 2014**), *Tg(fli1a:eGFP)[y1]* (**Lawson and Weinstein, 2002**), *Tg(kdrl:eGFP)[s843]* (**Jin et al., 2005**), *Tg(−5.2lyve1b:DsRed)[nz101]* (**Okuda et al., 2012**), and Zebrabow - *Tg(ubb:LOX2272-LOXP-RFP-LOX2272-CFP-LOXP-YFP)[a131]* (**Pan et al., 2013**). The *sox10*:CreERT2 transgene was generated with Gateway Cloning (Invitrogen) and the Tol2 kit (**Kwan et al., 2007**) by combining p5E-sox10 (**Das and Crump, 2012**), pME-CreERT2, p3E-pA, and pDestTol2pA2. The *col2a1a-R2-E1b*:CreERT2 and *col2a1a-R2-E1b*:H2A.F/Z-mCherry-P2A-GFPCAAX transgenes utilize a zebrafish *col2a1a* R2 enhancer element with a minimal E1B promoter sequence (**Dale and Topczewski, 2011**). For *col2a1a-R2-E1b*:CreERT2, CreERT2 was amplified using pENTR/D-CreERT2 as template and primers 5′- TTCTTGTACAAAG TGGCCACCGGCCACCATGTCCAATTTACTGACCGTACAC-3′ and 5′-TAGAGGCTCGAGAGGCC

TTGTCAAGCTGTGGCAGGGAAACCCTC-3'. The amplified PCR product was combined with an NcoI/EcoRI fragment of pDestTol2-col2a1aR2-E1B-eGFPpA (*Askary et al., 2015*) using Gibson Assembly (New England Biolabs). For *col2a1a-R2-E1b*:H2A.F/Z-mCherry-P2A-GFPCAAX, H2A.F/Z-mCherry was amplified using template pME-H2A.F/Z-mCherry and primers 5'-TTTCTTGTACAAAG TGGCCAAAGCTTGGATCCCGGCCACCATGGCAGGTGGAAAAGCAGG-3' and 5'-AAGTTGGTTGC TCCCGACCCCTTGTACAGCTCGTCCATGCCGCCGGTG-3'. P2A-GFPCAAX was synthesized as a gBlock (IDT). H2A.F/Z-mCherry and P2A-GFPCAAX were combined with an NcoI/EcoRI fragment of pDestTol2-col2a1aR2-E1B-eGFPpA using Gibson Assembly. All transgenes were injected at 30 ng/µL along with 50 ng/µL Tol2 mRNA into one-cell-stage embryos, with these animals raised and out-crossed to identify stable germline founders. CreERT2 transgenes were injected into *Tg(bactin2: loxP-BFP-loxP-DsRed)*[sd27] embryos, followed by overnight treatment with 10 µM (Z)−4-Hydroxyta-moxifen (4-OHT) (Sigma-Aldrich H7904) at two dpf and screening for DsRed+ chondrocytes at six dpf using a Leica fluorescent stereomicroscope. Embryos with fluorescent cartilages were raised to adulthood and outcrossed to identify founders. We used *Tg(sox10:CreERT2)*[el777], *Tg(col2a1a-R2-E1b:CreERT2)*[el712], and *Tg(col2a1a-R2-E1b:H2A.F/Z-mCherry-P2A-GFPCAAX)*[el695] lines for our experiments. Two additional alleles of *Tg(col2a1a-R2-E1b:CreERT2)* and one additional allele of *Tg (col2a1a-R2-E1b:H2A.F/Z-mCherry-P2A-GFPCAAX)* gave similar cartilage-specific expression. To generate *mmp9*[el734] we used CRISPR/Cas9 mutagenesis to target the second exon. gRNA targeting the sequence 5'-TTGATGCCATGAAGCAGCCC-3' was injected at 25 ng/µl with 50 ng/µl Cas9 RNA into one-cell-stage embryos as described (*Hwang et al., 2013*). The *mmp9*[el734] allele is an 8 bp dele-tion that results in the incorporation of 13 additional amino acids after amino acid 98 (P), followed by a premature stop codon that is predicted to completely abolish the catalytic metalloproteinase domain.

## Histology and LipidTOX staining

Live bone staining of dissected ceratohyal bones was performed by treating with 50 µg/ml Alizarin Red (Sigma Aldrich, cat. no. A5533), Calcein Green (Thermofisher Scientific, cat. no. C481), or Cal-cein Blue, AM (Thermofisher Scientific, cat. no. C1429) for 5 min and repeatedly rinsing in embryo medium as described (*Paul et al., 2016*). We performed adipocyte labeling of dissected ceratohyal bones by incubating in a 1:200 solution of HCS LipidTOX Deep Red (Life Technologies, cat. no. H34477) for 15 min and rinsing in embryo medium as described (*Minchin and Rawls, 2017*). Paraffin embedding and histology were performed as described (*Paul et al., 2016*). We cut blocks into 5 µm sections on a Shandon Finesse Me+ microtome (cat. no. 77500102) and collected sections on Apex Superior Adhesive slides (Leica Microsystems, cat. no. 3800080). Pentachrome and Trichrome stain-ing were performed according to manufacturer's instructions (Movat-Russell modified pentachrome stain kit, Newcomer Supply cat. no. 9150A; Gomori One-Step, Aniline Blue, trichrome stain kit, New-comer Supply cat. no. 9176A).

## Immunohistochemistry and in situ hybridization

For cell proliferations assays, fish were immersed in system water containing 4.5 mg/ml BrdU (Sigma Aldrich, cat. no. B5002) for 1 hr, followed by two washes in system water, fixation in 4% paraformal-dehyde, and paraffin embedding. Immunohistochemistry was performed as described except that we blocked with 2% normal goat serum (Jackson ImmunoResearch, cat. no. 005-000-121). After cut-ting thin sections, we performed antigen retrieval by treating slides with citrate buffer (pH 6.0) in a steamer set (IHC World, cat. no. IW-1102) for 35 min. Primary antibodies include rat anti-BrdU (1:100, Bio-Rad, cat. no. MCA2060GA) and rabbit anti-mCherry (1:250, Novus Biologicals, cat. no. NBP2-25157). We used AlexaFluor secondary antibodies and Hoechst to visualize nuclei. Colorimet-ric in situ hybridization was performed as described (*Paul et al., 2016*). The *mmp9* riboprobe was generated by PCR amplification of zebrafish genomic DNA with primers 5'-GTCTCCAATAC TAAAGCTCTGAAGAAG-3' and 5-'TAGGATGTCGAAGGTCTATAGAGAATG-3' and cloning into pCR-BluntII-TOPO (Life Technologies). RNA probe was synthesized with T7 polymerase (Roche) after linearizing the plasmid with BamHI restriction. RNAscope probes for *leptin receptor* (*lepr*) and *runx2b* were synthesized by Advanced Cell Diagnostics in Channel 1 and Channel 2, respectively, and for mouse *Lepr* in channel 1. Paraformaldehyde-fixed paraffin-embedded sections were

deparaffinized, and the RNAscope fluorescent multiplex v2 assay combined with immunofluorescence was performed according to manufacturer's protocols and with the ACD HybEZ Hybridization oven.

## Neural crest transplantations

At six hpf, donor ectoderm from the animal cap of *bactin2*:loxP-tagBFP-loxP-DsRed embryos was transplanted into the neural crest precursor domain of *mmp9*$^{-/-}$ hosts as described (*Crump et al., 2004*). Embryos displaying unilateral tagBFP fluorescence in the face at three dpf were raised in the nursery and then genotyped at 14 dpf for the *mmp9* mutant allele. Homozygous mutant and wild-type siblings were raised at similar density until they reached the indicated sizes for analysis.

## Imaging

Brightfield images of pentachrome and trichrome stains, and colorimetric in situ hybridizations, were acquired with a Zeiss AxioImager.A1 microscope and a Zeiss slide scanner AxioScan.Z1. Focus stacking of multiple images was done in Adobe Photoshop CS5. Fluorescence images were captured on a Zeiss LSM800 confocal microscope, with representative sections or maximum intensity projections shown as specified. Tiling was performed using ZEN software or manually stitched together using Fiji. Brightness and contrast were adjusted in Adobe Photoshop CS5 with similar settings for experimental and control samples.

## Quantification and statistical analyses

We stage-matched control and experimental zebrafish by measuring standard body length (SL) from the tip of the snout to the edge of the hypuralia. Adipocyte counts and area/width measurements of growth plates were calculated using Fiji. The proliferative zone of the growth plate was defined as the central region of compact chondrocytes. All measurements were performed blinded to genotype. Statistical significance was determined by a student's t-test for pair-wise comparisons or Tukey-Kramer HSD tests for multiple comparisons, using GraphPad's Prism software.

## Acknowledgments

We thank Megan Matsutani and Jennifer DeKoeyer Crump for fish care, Francesca Mariani for insightful conversations, David Traver for the *bactin2*:loxP-tagBFP-stop-loxP-DsRed line, and Ken Poss, Leonard Zon, and Rodney Dale for plasmids.

## Additional information

### Funding

| Funder | Grant reference number | Author |
|---|---|---|
| National Institute of Dental and Craniofacial Research | R35 DE027550 | J Gage Crump |
| Burroughs Wellcome Fund | Postdoctoral Fellowship | D'Juan T Farmer |
| National Institute of Dental and Craniofacial Research | F31 025549 | Dion Giovannone |
| National Institute of Dental and Craniofacial Research | K99 DE027218 | Joanna Smeeton |

The funders had no role in study design, data collection and interpretation, or the decision to submit the work for publication.

### Author contributions

Dion Giovannone, Data curation, Formal analysis, Investigation, Methodology, Writing—original draft, Writing—review and editing; Sandeep Paul, Conceptualization, Data curation, Formal analysis, Investigation, Methodology; Simone Schindler, Punam Patel, Data curation, Investigation, Methodology; Claire Arata, D'Juan T Farmer, Joanna Smeeton, Resources, Methodology; J Gage Crump,

Conceptualization, Supervision, Funding acquisition, Writing—original draft, Project administration, Writing—review and editing

## Author ORCIDs
Sandeep Paul (iD) https://orcid.org/0000-0002-2721-1874
J Gage Crump (iD) http://orcid.org/0000-0002-3209-0026

## Ethics

Animal experimentation: This study was performed in strict accordance with the recommendations in the Guide for the Care and Use of Laboratory Animals of the National Institutes of Health. All of the animals were handled according to approved institutional animal care and use committee (IACUC) protocols (#20771) of the University of Southern California.

## Decision letter and Author response

Decision letter https://doi.org/10.7554/eLife.42736.018
Author response https://doi.org/10.7554/eLife.42736.019

## Additional files

### Supplementary files

• Transparent reporting form
DOI: https://doi.org/10.7554/eLife.42736.016

### Data availability

All data generated or analysed during this study are included in the manuscript and supporting files.

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
