## [Decision Letter]

Thank you for submitting your article "Programmed Dedifferentiation of Hypertrophic Chondrocytes Generates Osteoblasts and Adipocytes in Zebrafish Bones" for consideration by *eLife*. Your article has been reviewed by three peer reviewers, and the evaluation has been overseen by a Reviewing Editor and Didier Stainier as the Senior Editor. The following individuals involved in review of your submission have agreed to reveal their identity: David Maridas (Reviewer #2).

The reviewers have discussed the reviews with one another and the Reviewing Editor has drafted this decision to help you prepare a revised submission.

Reviewing Editor: All agree the work is novel and represents potentially important differences with the more frequently studied mammalian systems. As such there is support for the manuscript. But there are several limitations that need to be addressed in a revision. First a clearer characterization of *LepR*+ cells is required since the authors conclusions differ from the kinetics in mice, raising an important distinction if true between fish and mammals. The comparisons between species however, are further limited by the difference in the *LepR* isoform used in the Cre lines between those of Morrison and in this work. Second, several components of Figure 3 are not clear and raise questions about labeling and timing of 4OHT- this aspect will require more experiments. Third, the results from the *MMP9* mutation are not directly related to the major tenet of the manuscript and the rescue experiments are based on small numbers of fish. Finally, it is conceptually difficult to appreciate the authors conclusions that the marrow adipocytes represent a fat reserve and that hematopoiesis follows later- there are no data to support that assumption. Are the authors suggesting the adipocyte depot represents a place holder for subsequent hematopoiesis?

As such these four points plus other comments from the reviewers will require more experiments and a comprehensive revision.

*Reviewer #1:*

This elegant and well written manuscript describes how chondrocytes in the zebrafish CH cartilage template gives rise to osteoblasts and adipocytes during CH bone maturation. The studies show that the CH cartilage template undergoes remodeling and vascularization during juvenile stages to generate fat-filled bones in the adult. Growth plate chondrocytes re-enter the cell cycle, and express the progenitor cell marker leptin receptor, suggesting dedifferentiation into progenitor cells, and then re-differentiation into osteoblasts, adipocytes and mesenchymal cells within the adult CH.

This study answers the important question as to whether zebrafish bone arises through invasion of the vasculature and conversion of growth plate cartilage to bone, as occurs in mammals. And also whether hypertrophic chondrocytes directly transform into osteoblast, or do so through a stem cell intermediate. This study identifies growth plate chondrocytes as a source of osteocytes and adipocytes in zebrafish bones, and suggest that dedifferentiation into proliferative progenitors as the likely mechanism.

The authors describe the bidirectional growth of the zebrafish CH growth plates (in contrast to mammalian unidirectional growth plates in long bones), and use beautiful lineage labeling studies to show chondrocyte clonal cell contributions to mesenchymal cells and adipocytes – and possibly osteoblasts – after growth plate remodeling.

Multiple approaches are used to demonstrate that hypertrophic chondrocytes and their osteoblast derivatives undergo cell division, indicating a dedifferentiation of hypertrophic chondrocytes. Conversion of hypertrophic chondrocytes to osteoblasts/adipocytes via transdifferentiation or through partial de-differentiation into a proliferative progenitor was examined using BrdU labeling. The expression of the progenitor cell marker *lepr* was used to monitor hypertrophic chondrocyte differentiation and de-differentiation into a progenitor cell-like state.

The authors conclude that growth plate remodeling is a conserved feature of bony vertebrates, and that chondrocyte dedifferentiation is a mechanism to generating osteoblasts and marrow adipocytes in the zebrafish. They also describe roles for *Mmp9*/13 in this process.

I recommend that this manuscript be accepted for publication in *eLife*.

Strengths:

1) Beautiful imaging, well described and careful studies.

2) The finding that late-stage hypertrophic chondrocytes give rise to osteoblasts and marrow adipocytes in the zebrafish CH.

3) Chondrocytes also give rise to both endosteal and periosteal surfaces fo the Ch bone, similar to mouse.

4) Demonstration that *mmp9* is required, in the neural crest cell lineage and perhaps in chondrocytes themselves, for Ch growth plate width, remodeling and adipocyte number, using both *mmp9* mutant and rescue by WT cell transplantation into *mmp9* mutant zebrafish.

5) The finding that growth plate remodeling is well conserved between mammals and fishes.

6) Use of three different methods of lineage tracing, including a Cre independent technique, to show that late-stage hypertrophic chondrocytes generate not only osteoblasts but also marrow adipocytes in zebrafish.

7) The interesting hypothesis that evolutionarily, fish bone marrow may represent the ancestral condition of being primarily a fat reservoir, with hematopoiesis moving into the bone later during tetrapod evolution.

8) The intriguing hypothesis that *mmp9* may have additional non-canonical function in the nucleus for histone H3 tail cleavage in hypertrophic chondrocytes, as has been demonstrated for osteoclasts.

9) Discrepancies with the mouse *lepr* expression patterns is discussed, and proposed that this may be due to the more restricted expression of the long *LepR* isoform (mouse) as compared to the short isoform. Direct analyses of the two isoforms in zebrafish is proposed.

*Reviewer #2:*

This manuscript elegantly demonstrates the significant contribution of hypertrophic chondrocytes to bone marrow fat in zebrafish. The authors also provide solid data to back their conclusions and suggest a novel mechanism of de-differentiation.

However, I have two comments about the context and the data presented in this article:

1) Although they mentioned the possibility that chondrocytes differentiate into adipocytes in mammals in the discussion, the authors missed on placing their findings with the current literature in mammals, notably mouse studies. Murine chondrocytes cultures can differentiate in adipocytes in vitro. Also, Ono et al., 2014, have previously showed that Col2CRE-ER dtomato mice do not show adipocyte labelling at 1 week, 1month or even 1 year after tamoxifen induction. It would be interesting for the authors to mention whether this specific to fish or not.

2) Would it be possible for the authors to quantify the contribution of chondrocytes to the total number of adipocytes? In other words, what is the ratio of labelled vs non-labelled adipocytes in their adult animals?

3) Is it possible that WT donor cells coming from WT did not rescue the number of adipocytes in *mmp9* mutants because of the low n in that experiment (i.e. n = 2)? If so, the authors should mention the limitation of a low n.

*Reviewer #3:*

In this manuscript, Giovannone et al. examine the results of neural crest and cartilage lineage tracing in a zebrafish model, that chondrocytes can contribute to mesenchyme in the endosteal compartment. Overall, the observations made here mostly mirror previously published studies in mouse systems. While there are novel insights to be gained from further lineage tracing studies in a zebrafish system to understand if these findings are conserved, the existence of extensive prior literature utilizing this approach in mice generally limits the additional novel insights to be gained unless the findings of this system are pursued in substantial depth, which is largely not the case in the present manuscript.

A few potentially significant novel observations are made but not followed up upon. For example, the labeling with the *Col2a1* system performed here is very restricted compared with use of similar reagents in mice. This is of interest for suggesting that the presence of chondrocyte-like gene expression signatures in bona fide MSCs is restricted to mammalian systems and therefore a recent evolutionary development. However, this is not explored in sufficient detail to flesh out this possibility and draw clear conclusions on this point. Similarly potentially interesting observations are made regarding the origin of *LEPR*+ cells, however these are not explored in sufficient detail to provide clear answers as to the origin of marrow resident *LEPR* cells. Additional specific comments are described below:

Major points:

1) It is unclear if the timing of 4-OHT administration truly restricts Sox10-cre ERT driven recombination to cartilage and not to other populations of neural crest-derived progenitors of the Ch bone. Indeed, the labeling of the periosteum observed suggests that the latter possibility is more likely. Given that Sox10 may label neural crest-derived mesenchymal progenitors beyond just chondrocytes, the use of Sox10-based lineage tracking is problematic for experiments designed to inform on chondrocyte conversion into osteoblast or adipocyte-lineage cells. The comparison to the broader deletion mediated by the human Sox10-cre is helpful, but does not exclude labeling of other non-cartilage MSC populations. It is appreciated that the *Col2a1* system addresses some of these weaknesses of the Sox10 system.

2) The images shown in Figure 3E are not convincing that they represent solely labeling of 3 independent clones as the regions of the distinct clones appear to partially overlap in the image available, undercutting the claim of clonal labeling. Titration studies of 4-OHT are needed to establish the dose ranges where clonal labeling is observed. Moreover even with full validation of a dose titration series, it is impossible to be certain that labeling represents true single clone derivatives unless either serial imaging is performed or a multiclone reporter similar to the confetti mouse is used. The suggestion that *col2a1*-mediated labeling is restricted in zebrafish relative to observations made in mice is interesting for suggesting that the presence of a chondrocyte-like gene expression program in MSC populations is restricted to mammals. However, further experiments to directly assess this would be needed to flesh out this possibility.

3) A major limitation of this study is that it only tracks cells but does not assess their functional contribution to mineralizaiton, for instance by deleting a gene required for mineralization using *Col2a1*-cre in zebrafish and observing the actual phenotypic contribution of post-chondrocyte osteoblasts to skeletal mineralization.

4) The investigation of cartilage cells as a source of *LEPR*+ cells is primarily of interest for the suggestion that cartilage is the source of the *LEPR*+ cell populations described in mice. However, these part of the manuscript has limitations that prevent it from providing substantial insights into the question of the source of the *LEPR*+ cells as they are described in the murine system.

Aside from these concerns, one of the notable features of murine *LEPR*-cre marked cells is that *LEPR* generally excludes cells in the growth plate. Thus, the significance of *LEPR* labeling in cartilage in fish for mammalian physiology is unclear. Indeed, the labeling in cartilage appears quite weak relative to the labeling in cells adjacent to bone and it is unclear what degree of signal in this in situ hybridization based system would correspond to the cells marked by *LEPR*-cre in mice. There is also no characterization of the *LEPR*+ cells offered, so it is unclear to what degree these cells observed in zebrafish correspond functionally to murine *LEPR*+ cells. Additionally, a defining feature of murine *LEPR* osteogenic cells is that they do not emerge until adulthood. What are the kinetics of *LEPR*+ cells emerging during the lifetime of the fish?

5) The work on *MMP9* is not strongly related to the rest of the manuscript. No direct evidence is offered to show that any of the phenotypes seen with the *MMP9* loss of function relate to transdifferentiation of cartilage cells.

6) In Figure 6, how can it be excluded that the Brdu labeling observed is due to labeling of proliferating chondrocytes that have progressed to the hypertrophic stage by the time of visualization?

---

## [Author Response]

Reviewing Editor: All agree the work is novel and represents potentially important differences with the more frequently studied mammalian systems. As such there is support for the manuscript. But there are several limitations that need to be addressed in a revision.First a clearer characterization of LepR+ cells is required since the authors conclusions differ from the kinetics in mice, raising an important distinction if true between fish and mammals.

In a new Figure 6—figure supplement 1, we have performed in situ analysis of *lepr* at 4 additional stages in zebrafish, as well as in situ analysis of *Lepr* in the postnatal mouse femur. These experiments show that *lepr*+ cells in the fish marrow emerge only after remodeling of the growth plate. We also find similar expression of fish *lepr* and mouse *Lepr* in chondrocytes at all stages, with expression in both species most prominent in the proliferative zone of the growth plate. We reference previous papers showing similar chondrocyte expression of *Lepr* in mouse, rat, and human, and that the marrow-specific expression of mouse *Lepr-Cre* is likely due to this transgene targeting only the long isoform of *Lepr*. In response to reviewer 3, we have modified the Discussion to stress that our results show that *lepr*+ marrow cells are derived from chondrocytes, and that further experiments are needed to determine if these are equivalent to the *Lepr-Cre*+ marrow stem cells reported in mouse.

Second, several components of Figure 3 are not clear and raise questions about labeling and timing of 4OHT- this aspect will require more experiments.

We agree with reviewer 3 that our initial analysis using *sox10*:CreER was problematic given its expression in additional non-cartilage cells. In a new Figure 3F, we have now performed more precise clonal labeling using zebrabow fish as requested, which reveals common color clones of chondrocytes and adipocytes, and chondrocytes and putative osteocytes. This strengthens our argument that it is *sox10*:CreER-labeled chondrocytes that give rise to adipocytes and osteocytes.See response to reviewer 3 for more details.

Third, the results from the MMP9 mutation are not directly related to the major tenet of the manuscript and the rescue experiments are based on small numbers of fish.

We disagree that the results from the *mmp9* mutant analysis are not directly related to the manuscript. The *mmp9* mutant analysis is used to make two major points. First, it was previously unclear whether fish growth plates undergo the same type of remodeling seen in mammalian bones. A common delay in growth plate remodeling in fish *mmp9* and mouse *Mmp9* mutants therefore supports genetic conservation of remodeling in both species. Second, we show that zebrafish *mmp9* functions in chondrocytes themselves for remodeling. By increasing our *n* number for rescue experiments (from *n=2* to *n=8* in the revision) we now see that restoring *mmp9* in chondrocytes rescuesadipocyte generation (revised Figure 7G). Thus, analysis of *mmp9* mutants provides important genetic evidence supporting our major tenet that remodeling of chondrocytes generates adipocytes.

Finally, it is conceptually difficult to appreciate the authors conclusions that the marrow adipocytes represent a fat reserve and that hematopoiesis follows later- there are no data to support that assumption. Are the authors suggesting the adipocyte depot represents a place holder for subsequent hematopoiesis?

We acknowledge that this evolutionary statement was highly speculative and have therefore removed it from the Discussion.

As such these four points plus other comments from the reviewers will require more experiments and a comprehensive revision.

Reviewer #2:

This manuscript elegantly demonstrates the significant contribution of hypertrophic chondrocytes to bone marrow fat in zebrafish. The authors also provide solid data to back their conclusions and suggest a novel mechanism of de-differentiation.However, I have two comments about the context and the data presented in this article:

*1) Although they mentioned the possibility that chondrocytes differentiate into adipocytes in mammals in the discussion, the authors missed on placing their findings with the current literature in mammals, notably mouse studies. Murine chondrocytes cultures can differentiate in adipocytes* in vitro*. Also, Ono et al., 2014, have previously showed that Col2CRE-ER dtomato mice do not show adipocyte labelling at 1 week, 1month or even 1 year after tamoxifen induction. It would be interesting for the authors to mention whether this specific to fish or not.*

We thank the reviewer for alerting us to these previous studies. A re-examination of (Ono et al., 2014) revealed that they did in fact observe adipocyte labeling post-chase –“During an extended chase period over a year after the pulse, Col2^creER^-P3 cells continued to yield chondrocytes, osteoblasts and stromal cells in the metaphysis, and also became adipocytes in the metaphyseal bone marrow (Figure 3C,D and Supplementary Figure 2F)”. We now include a discussion of the current mouse literature.

“Indeed, older studies have demonstrated the ability of murine chondrocytes to differentiate into adipocytes in vitro (Heermeier et al., 1994; Hegert et al., 2002). Further, *Col2a1*:CreER cells converted at postnatal day 3 in mouse were found to give rise to adipocytes in the metaphyseal bone marrow, although a caveat is that *Col2a1*:CreER marks both chondrocytes and progenitors at early postnatal stages (Ono et al., 2014).”

2) Would it be possible for the authors to quantify the contribution of chondrocytes to the total number of adipocytes? In other words, what is the ratio of labelled vs non-labelled adipocytes in their adult animals?

We considered quantification but concluded that it would not be skewed due to several factors. First, conversion efficiency is variable in our CreER lines. Second, perhaps due to gene silencing or the small amount of cytoplasm in mature adipocytes, we cannot always definitively detect the reporter allele (converted or unconverted) in all adipocytes. Thus, while we could certainly attempt to quantify the proportion of lineage cells, any quantification would be an under-representation of chondrocyte contribution to adipocytes.

3) Is it possible that WT donor cells coming from WT did not rescue the number of adipocytes in mmp9 mutants because of the low n in that experiment (i.e. n = 2)? If so, the authors should mention the limitation of a low n.

We have now performed additional transplantation experiments and have increased our n from 2 to 8, which reveal a strong trend toward rescue of adipocyte numbers (revised Figure 7G).

“We also observed rescue of marrow adipocyte number by wild-type chondrocytes, though this had moderate statistical significance (p = 0.06), potentially owing to low sample size (*n* = 8), the mosaic contribution of wild-type cells to the growth plate, and/or roles of cells outside the growth plate to marrow adipocyte generation.”

Reviewer #3:

In this manuscript, Giovannone et al. examine the results of neural crest and cartilage lineage tracing in a zebrafish model, that chondrocytes can contribute to mesenchyme in the endosteal compartment. Overall, the observations made here mostly mirror previously published studies in mouse systems. While there are novel insights to be gained from further lineage tracing studies in a zebrafish system to understand if these findings are conserved, the existence of extensive prior literature utilizing this approach in mice generally limits the additional novel insights to be gained unless the findings of this system are pursued in substantial depth, which is largely not the case in the present manuscript.A few potentially significant novel observations are made but not followed up upon. For example, the labeling with the Col2a1 system performed here is very restricted compared with use of similar reagents in mice. This is of interest for suggesting that the presence of chondrocyte-like gene expression signatures in bona fide MSCs is restricted to mammalian systems and therefore a recent evolutionary development. However, this is not explored in sufficient detail to flesh out this possibility and draw clear conclusions on this point. Similarly potentially interesting observations are made regarding the origin of LEPR+ cells, however these are not explored in sufficient detail to provide clear answers as to the origin of marrow resident LEPR cells. Additional specific comments are described below:

We thank the reviewer for this comment and agree that we had not previously explained key differences in the way the zebrafish *col2a1a*-CreER line was constructed compared to its equivalent in mouse. We now clarify that the col2a1a-CreER transgene was made with a cartilage-specific enhancer that we have extensively validated in a previous study (Askary et al., 2015). This avoids the weak expression of zebrafish *col2a1a* in perichondrium/periosteum, which we now reference. In the future, a zebrafish *col2a1a*-CreER knock-in line that reports full *col2a1a* expression (including in the perichondrium/periosteum) would be needed to test whether *col2a1a*+ MSCs similar to those reported by (Ono et al., 2014) also exist in zebrafish.

“In mice, *Col2a1*:CreERT2-mediated conversion at embryonic and early postnatal stages broadly labels not only chondrocytes but also osteochondroprogenitors, such as those in the perichondrium and periosteum (Ono et al., 2014). […]Treatment of *col2a1a/B>R* fish with a single dose of 4-OHT at 5 dpf resulted in extensive labeling of *col2a1a-BAC*:GFP+chondrocytes at 12 dpf, but no labeling of the perichondrium, periosteum, and osteoblasts as marked by the *sp7*:GFP transgene (Figure 4A, B; Figure 3—figure supplement 1B).”

In a new Figure 4A,B, we present additional controls (including high magnification views of the perichondrium) showing co-localization of col2a1a-CreER-converted cells with the chondrocyte marker *col2a1a-BAC*:GFP but not the osteoblast and perichondrium/periosteum marker *sp7*:GFP. These additional controls further validate the use of this CreER line to follow the long-term fates of fish chondrocytes in a specific manner.

Major points:1) It is unclear if the timing of 4-OHT administration truly restricts Sox10-cre ERT driven recombination to cartilage and not to other populations of neural crest-derived progenitors of the Ch bone. Indeed, the labeling of the periosteum observed suggests that the latter possibility is more likely. Given that Sox10 may label neural crest-derived mesenchymal progenitors beyond just chondrocytes, the use of Sox10-based lineage tracking is problematic for experiments designed to inform on chondrocyte conversion into osteoblast or adipocyte-lineage cells. The comparison to the broader deletion mediated by the human Sox10-cre is helpful, but does not exclude labeling of other non-cartilage MSC populations. It is appreciated that the Col2a1 system addresses some of these weaknesses of the Sox10 system.2) The images shown in Figure 3E are not convincing that they represent solely labeling of 3 independent clones as the regions of the distinct clones appear to partially overlap in the image available, undercutting the claim of clonal labeling. Titration studies of 4-OHT are needed to establish the dose ranges where clonal labeling is observed. Moreover even with full validation of a dose titration series, it is impossible to be certain that labeling represents true single clone derivatives unless either serial imaging is performed or a multiclone reporter similar to the confetti mouse is used. The suggestion that col2a1-mediated labeling is restricted in zebrafish relative to observations made in mice is interesting for suggesting that the presence of a chondrocyte-like gene expression program in MSC populations is restricted to mammals. However, further experiments to directly assess this would be needed to flesh out this possibility.

We agree with the reviewer about the limitations of using *sox10*:CreER given its additional activity outside chondrocytes (e.g. in the perichondrium). In our hands, conversion with 4OH-T at 14 dpf gave the most robust conversion in cartilage, and we have yet to find a time of 4OH-T treatment that only activates in cartilage. We also acknowledgethe limitations of our original clonal analysis given the overlapping nature of the clones. We have therefore performed new experiments using the multicolor zebrabow reporter, which operates similarly to the confetti mouse. New data in Figure 3F now more clearly support clonal contribution of chondrocytes to adipocytes and putative osteocytes.

“To more definitively follow growth plate clones, we also examined *sox10*:CreERT2; *ubb*:Zebrabow fish in which 4OH-T treatment at 14 dpf resulted in conversion of RFP (red) to various color combinations of CFP, YFP, and RFP at 23 mm SL. Analysis of uniquely colored clones in the growth plate revealed those that contained both chondrocytes and adjacent adipocytes in the marrow, as well as those that contained chondrocytes and cells embedded in cortical bone, consistent with an osteoblast/osteocyte identity (Figure 3F).”

3) A major limitation of this study is that it only tracks cells but does not assess their functional contribution to mineralizaiton, for instance by deleting a gene required for mineralization using Col2a1-cre in zebrafish and observing the actual phenotypic contribution of post-chondrocyte osteoblasts to skeletal mineralization.

We agree that this would be an informative experiment in the future. Unfortunately at present, precise integration of loxP sites into the zebrafish genome is technically very challenging. To our knowledge, there is only one report to date claiming to have accomplished this (lab of David Grunwald). Even if possible, it would be a lengthy process constructing these fish, and as such we feel attempting this would be well beyond the scope of the current study.

4) The investigation of cartilage cells as a source of LEPR+ cells is primarily of interest for the suggestion that cartilage is the source of the LEPR+ cell populations described in mice. However, these part of the manuscript has limitations that prevent it from providing substantial insights into the question of the source of the LEPR+ cells as they are described in the murine system.Aside from these concerns, one of the notable features of murine LEPR-cre marked cells is that LEPR generally excludes cells in the growth plate. Thus, the significance of LEPR labeling in cartilage in fish for mammalian physiology is unclear. Indeed, the labeling in cartilage appears quite weak relative to the labeling in cells adjacent to bone and it is unclear what degree of signal in this in situ hybridization based system would correspond to the cells marked by LEPR-cre in mice. There is also no characterization of the LEPR+ cells offered, so it is unclear to what degree these cells observed in zebrafish correspond functionally to murine LEPR+ cells. Additionally, a defining feature of murine LEPR osteogenic cells is that they do not emerge until adulthood. What are the kinetics of LEPR+ cells emerging during the lifetime of the fish?

In a new Figure 6—figure supplement 1, we perform *lepr* in situs at 4 additional stages in zebrafish, as well as *Lepr* in situs in mouse. These new data show conserved expression of fish/mouse *Lepr* in growth plate chondrocytes, with strongest expression in the proliferative zone. They also reveal emergence of *lepr*+ marrow cells in zebrafish only after initiation of growth plate remodeling (i.e. at 15, 20, 27 mm stages, but not at 8 and 12 mm). We discuss that this likely reflects the *Lepr*-Cre line only reporting expression of the long-isoform version of *Lepr*.

“One caveat is that we detect endogenous *lepr* expression in both marrow cells and growth plate chondrocytes in zebrafish. However, the more specific labeling of marrow cells by the *LepR*-Cre in mouse likely reflects the Cre insertion being in the long *LepR* isoform (containing exon 18b) that displays more restricted expression than the short isoform (Zhou, BO et al., 2014). Indeed, *LepR* mRNA and protein has also been reported in chondrocytes of mouse (Hoggard et al., 1997), rat, and human (Morroni et al., 2004), a finding we confirmed in the postnatal mouse femur, including the same higher expression in immature versus hypertrophic chondrocytes that we observe in zebrafish (Figure 6—figure supplement 1B).”

LEPR does not in and of itself specify a skeletal stem cell as mentioned in the text. Rather it labels a heterogeneous population of cells (as shown in Zhou BO et al. 2014) that may include a stem cell subset, though this has yet to be directly addressed experimentally.

We have revised the text to provide a more conservative treatment of LEPR as suggested.

“In mice, *LepR* expression marks a heterogeneous population of cells in endochondral bone, including a putative postnatal skeletal stem cell population (Zhou, BO et al., 2014)… These results are consistent with *lepr*+ cells in the bone marrow deriving from growth plate chondrocytes in zebrafish, although direct evidence will be needed to determine if any of these chondrocyte-derived *lepr+* marrow cells behave as skeletal stem cells in zebrafish.”

“Without a comparable long-isoform *lepr*-Cre line in zebrafish, we cannot therefore conclude whether zebrafish *lepr*+ marrow cells are comparable to those described in mouse. In the future, the generation of new Cre lines to specifically mark *lepr*+ marrow cells in zebrafish will be needed to determine whether these chondrocyte-derived marrow cells also act as stem cells for osteoblasts and adipocytes in post-embryonic fish.”

5) The work on MMP9 is not strongly related to the rest of the manuscript. No direct evidence is offered to show that any of the phenotypes seen with the MMP9 loss of function relate to transdifferentiation of cartilage cells.

We feel that the *mmp9* mutant data make several points of interest that relate to the manuscript. First, it establishes a similar genetic dependency of growth plate remodeling in zebrafish and mouse. Prior to our work, it was unclear whether mammalian-like growth plate remodeling operated in fish. Combined with the lineage tracing data presented, the similar *mmp9* phenotypes further support conservation. In this revision, we have also performed additional *mmp9* rescue experiments (increasing *n* from 2 to 8), which allow us to conclude that Mmp9 likely functions in chondrocytes for marrow fat production, thus providing additional supporting evidence for the conversion of chondrocytes into adipocytes (see response to reviewer 2 for caveats to this analysis in the revised Discussion).

6) In Figure 6, how can it be excluded that the Brdu labeling observed is due to labeling of proliferating chondrocytes that have progressed to the hypertrophic stage by the time of visualization?

We explain in the Materials and methods that fish were treated with BrdU for 1 hour followed by immediate fixation, hence detecting only cells that are actively proliferating or have only very recently divided. Progression of chondrocytes from the proliferative to the hypertrophic stage occurs on the order of days.